

# Research and experimental verification on the mechanisms of cellular senescence in triple-negative breast cancer

Tengfei Cao[1,*], Mengjie Huang[1,*], Xinyue Huang[1] and Tian Tang[2]

[1] Department of Breast Surgery, The Second Affiliated Hospital of Guangzhou Medical University, Guangzhou, China

[2] Department of Pathology, The Second Affiliated Hospital of Guangzhou Medical University, Guangzhou, China

[*] These authors contributed equally to this work.

Corresponding author
Tian Tang, tt9772888@126.com

## ABSTRACT

**Background.** Triple-negative breast cancer (TNBC) is an aggressive breast cancer subtype with high heterogeneity, poor prognosis, and a low 10-year survival rate of less than 50%. Although cellular senescence displays extensive effects on cancer, the comprehensions of cellular senescence-related characteristics in TNBC patients remains obscure.

**Method.** Single-cell RNA sequencing (scRNA-seq) data were analyzed by Seurat package. Scores for cellular senescence-related pathways were computed by single-sample gene set enrichment analysis (ssGSEA). Subsequently, unsupervised consensus clustering was performed for molecular cluster identification. Immune scores of patients in The Cancer Genome Atlas (TCGA) dataset and associated immune cell scores were calculated using Estimation of STromal and Immune cells in MAlignantTumours using Expression data (ESTIMATE) and Microenvironment Cell Populations-counter (MCP-counter), Tumor Immune Estimation Resource (TIMER) and Estimating the Proportion of Immune and Cancer cells (EPIC) methods, respectively. Immunotherapy scores were assessed using TIDE. Furthermore, feature genes were identified by univariate Cox and Least Absolute Shrinkage and Selection Operator (LASSO) regression analyses; these were used to construct a risk model. Additionally, quantitative reverse transcription-polymerase chain reaction (qRT-PCR) and transwell assay were conducted for *in vitro* validation of hub genes.

**Result.** TNBC was classified into three subtypes based on cellular senescence-related pathways as clusters 1, 2, and 3. Specifically, cluster 1 showed the best prognosis, followed by cluster 2 and cluster 3. The levels of gene expression in cluster 2 were the lowest, whereas these were the highest in cluster 3. Moreover, clusters 1 and 3 showed a high degree of immune infiltration. TIDE scores were higher for cluster 3, suggesting that immune escape was more likely in patients with the cluster 3 subtype who were less likely to benefit from immunotherapy. Next, the TNBC risk model was constructed and validated. RT-qPCR revealed that prognostic risk genes (*MMP28*, *ACP5* and *KRT6A*) were up-regulated while protective genes (CT83) were down-regulated in TNBC cell lines, validating the results of the bioinformatics analysis. Meanwhile, cellular experiments revealed that *ACP5* could promote the migration and invasion abilities in two TNBC cell lines. Finally, we evaluated the validity of prognostic models for assessing TME characteristics and TNBC chemotherapy response.

**Conclusion**. In conclusion, these findings help to assess the efficacy of targeted therapies in patients with different molecular subtypes, have practical applications for subtype-specific treatment of TNBC patients, and provide information on prognostic factors, as well as guidance for the revelation of the molecular mechanisms by which senescence-associated genes influence TNBC progression.

## INTRODUCTION

Triple-negative breast cancer (TNBC), an aggressive breast cancer subtype, lacks the expression of the progesterone receptor, human epidermal growth factor receptor 2 (HER2), and estrogen receptor (*Shang & Xu, 2022*). As a highly heterogeneous tumor, TNBC shows distinct biological features of aggressiveness, high recurrence rate, and distant metastasis (*Gluz et al., 2009*; *Foulkes, Smith & Reis-Filho, 2010*), which can be attributed to germline alterations (BRCA1 mutations), differences in genetic characteristics, epigenetic alterations, DNA repair defects, morphological features, and gene expression profiles (*Nik-Zainal et al., 2016*; *Jiang et al., 2019*; *Bareche et al., 2018*; *Pareja & Reis-Filho, 2018*). Due to this heterogeneity, large tumors may contain multiple cells with different molecular characteristics and displaying different sensitivity to treatment (*Dagogo-Jack & Shaw, 2018*), which has been demonstrated to be the main reason for drug resistance or lower efficiency of immunotherapy in breast cancer therapy (*Li et al., 2021*). Relative to other subtypes, TNBC has a poor prognosis with a low 10-year survival rate of less than 50% (*Malorni et al., 2012*). Effective therapeutic targets for TNBC are scarce (*Huang et al., 2021*; *Zhu, Yang & Xu, 2022*). Therefore, it is particularly crucial to assess the mechanisms underlying the progression and onset of TNBC for immunotherapy and its diagnosis in clinical settings.

The main feature of cellular senescence is a complex secretory phenotype, known as the senescence-associated secretory proteome (SASP), that by facilitate immune-mediated clearance of senescent cells to keep tissue homeostasis and suppresses tumorigenesis, as a powerful tumor-suppressing mechanism (*Bartkova et al., 2006*; *Xue et al., 2007*). For example, excessive reactiveoxygenspecies caused by metabolic perturbation could trigger mitogen-activated protein kinase (MAPK)/p38 pathways to facilitate cellular senescence (*Borodkina et al., 2016*). Researches also uncovered that the MEK/MAPK signaling pathway was involved in inducing senescence in human fibroblasts (*Takasugi, Yoshida & Ohtani, 2022*). Previous studies suggested that cancer cells have devised various ways to evade this mechanism (*Hollstein et al., 1991*; *Jarrard et al., 1999*; *Foster et al., 1998*). When co-transplanted with completely malignant cells, the growth of senescent cells increases along with the tumor formation rates in the xenografts (*Bartholomew, Volonte & Galbiati, 2009*; *Bhatia et al., 2008*; *Liu & Hornsby, 2007*). The impact of senescent cells on the evolution of precancerous tumor progression has been confirmed (*Campisi et al., 2011*; *Collado, Blasco & Serrano, 2007*). Proliferation is induced in several precancerous cells in

the presence of senescent fibroblasts, ultimately resulting in tumor formation (*Krtolica et al., 2001*; *Bavik et al., 2006*; *Castro-Vega et al., 2015*). On the one hand, cellular senescence, a fail-safe mechanism, inhibits tumorigenesis in a cell-autonomous environment by preventing harmful cells from further proliferation. On the other hand, senescent cells secrete excess growth factors and cytokines (*Coppé et al., 2010*), including interleukin 8 (IL-6) and IL-6 (collectively referred to as SASP) (*Lasry & Ben-Neriah, 2015*), thus establishing a microenvironment of immunosuppression, inflammation, and catabolism to promote chemotherapeutic resistance and tumor growth (*Coppé et al., 2008*). Intriguingly, within multiple element causing TNBC chemoresistance, therapy-induced senescence is well-accepted (*Chakrabarty et al., 2021*). Despite these extensive effects of cellular senescence on cancer, our understanding of cellular senescence-related characteristics of TNBC patients remains rudimentary. In particular, whether these characteristics of cancer patients can be used as biomarkers to guide clinical prognosis and treatment of TNBC remains elusive.

Given that cellular senescence signature is highly correlated with the progression of cancer, an increasing number of studies have investigated it and demonstrated that senescence related signature is significantly associated with the prognosis of tumor patients. For example, in lung adenocarcinoma (LUAD), a comprehensive analysis constructed a senescence-related signature score (SRS), which was elucidated as an independent prognostic factor for LUAD patients and revealed that it was positively associated with cancer-associated fibroblasts, NK cell infiltration, and cytokine release (*Lin et al., 2021*). Among them, cancer-associated fibroblasts often present a senescence-associated secretory phenotype (SASP) under stress conditions such as inflammatory factors, which promotes tumor formation and chemoresistance through the release of cytokine IL-6, which also reveals the relevant mechanism of cellular senescence-associated cancer progression from the level of regulation of the immune microenvironment (*Yasuda et al., 2021*; *Wu et al., 2017*). Another study revealed that SRS in colorectal cancer can be used to assess the genomic mutation status of the tissues, especially the high SRS group showed significant RPN2 mutations (*Yue et al., 2021*). RPN2 is commonly associated with cell proliferation, migration, and epithelial-mesenchymal transition phenotypes, exacerbates malignant cancer progression through the release of factors such as MMP-9, and inhibits cellular autophagy through STAT3 and NF-κB pathways (*Huang et al., 2019*; *Zhang et al., 2019*; *Han et al., 2023*). Although the clinical value of cellular senescence-related features has not been explored in TNBC, the above reports still inform our study, and we hypothesize that markers associated with senescence features and prognosis in TNBC could influence cancer progression and prognosis in terms of tumor infiltration microenvironment homeostasis, cytokine release, and genomic mutations.

Based on scRNA-seq and bulk RNA-seq data, we constructed a senescence classifier for TNBC based on cellular senescence-related pathways. Significant differences in prognosis, tumor microenvironment (TME), genome mutations and enriched metabolic pathways were detected from the three subtypes. Next, the heterogeneity of malignant cells among senescent and normal cells was further analyzed in TNBC. A cellular senescence-related risk model for TNBC was constructed. *In vitro* experiments were also conducted to confirm the expression levels of hub genes selected for the risk model and the functions

of certain gene on TNBC progression. Additionally, we also verified the reliability of the signature in assessing the regulatory role of the TNBC immune microenvironment and the patient's response to immunotherapy. Finally, the possibility of cellular senescence as a clinicopathological feature to improve the prognosis and survival prediction of patients with TNBC was explored. Collectively, our study provides a new direction to the diagnosis and treatment of patients with TNBC.

## MATERIAL AND METHODS

### Data downloading and pre-processing

Single-cell sequencing dataset, GSE176078, comprising nine samples was extracted from the NCBI data Gene Expression Omnibus (GEO; https://www.ncbi.nlm.nih.gov/geo/). The statistical power (*Therneau, Hart & Kocher, 2023*) of this experimental design, calculated in RNASeqPower is 1.0. Single-cell data were subjected to filtering, quality control, clustering, and dimensionality reduction using the "Seurat" (*Gribov et al., 2010*) package. Microarray data of GSE58812 was extracted from GEO, and probes were transformed to "symbols" using the annotation file. In total, 107 qualified tumor samples and 16,416 genes were obtained after excluding tumor samples and normal tissues without information on clinical follow-up and overall survival (OS).

Clinical phenotypic data of TNBC were extracted from The Cancer Genome Atlas (TCGA) database. Samples without information on survival time and statuses were removed, and all those with patient survival time >0 days were retained. Subsequently, expression profile data were downloaded from TCGA. Finally, 113 para-cancerous and 113 tumor samples were obtained. The statistical power of this experimental design, calculated in RNASeqPower is 1.0. Furthermore, copy number variations (CNVs) on the Masked Copy Number Segment type of TNBC were obtained from TCGA database and integrated using "gistic2" (*Mermel et al., 2011*) software. Finally, the single nucleotide variants (SNVs) in TCGA-TNBC dataset processed by "mutect2" software, were obtained from TCGA database. Cellular senescence-related pathways were extracted from the Molecular Signature Database (MSigDB: https://www.gsea-msigdb.org/gsea/index.jsp). Sangerbox platform (http://vip.sangerbox.com/) was introduced to assist bioinformatics analysis in this study (*Shen et al., 2022*).

### Single-cell clustering and dimensionality reduction

First, 38,582 cells were obtained after filtering the single-cell RNA-seq data matrix according to the cell criterion (<250 transcripts/cell) and gene criterion (<3 cells/gene). The "PercentageFeatureSet" function was then used to determine the percentages of mitochondria and rRNAs. Ultimately, 38,007 cells that met the following inclusion criteria were obtained: One cell could express between 100 and 6,000 genes, the mitochondrial content was less than 25%, and each cell had a unique molecular identifier (*Yasuda et al., 2021*) of at least 100.

Variance stabilization transformation (VST) was used to identify variable characteristics after the aforementioned data were log-normalized. High-variance genes were then discovered by employing the "FindVariableFeatures" tool. Further, the "FindIntegrationAnchors"

function was utilized to remove batch effects using the canonical correspondence analysis (CCA) method for 10 samples. The data were integrated using the "IntegrateData" function. The anchor (dim = 30) was determined by principal component analysis (PCA) and dimensionality reduction after normalization using the "ScaleData" function for all genes. The cells were clustered using "FindNeighbors" and "FindClusters" functions with resolution = 0.1; this yielded 11 subpopulations. Finally, the TSNE dimensionality reduction analysis was performed on 38,582 cells using the "RunTSNE" function.

With threshold parameters of logfc = 0.5 (fold-change), Minpct = 0.35 (minimum percentage of differential gene expression), and adjusted P0.05, marker genes of subpopulations were screened using the "FindAllMarkers" tool. Subsequently, changes in cellular CNVs in single-cell data were predicted using the "copycat" (*Gao et al., 2021*) package to distinguish tumor cells. Although cancerous and normal tissues were differentiated during sampling, cancerous tissues might contain normal cells. Therefore, a distinction between the two was necessary.

## Gene set enrichment analysis (GSEA) and annotation

Cellular senescence-related pathways were obtained from the GSEA website (https://www.gsea-msigdb.org/gsea/index.jsp). We separately calculated the corresponding scores of non-malignant and malignant cells in cellular senescence-related pathways by ssGSEA using the "GSVA" package and normalized the enrichment scores for each pathway by z-score. The scores of each sample of normal and tumor tissues in the bulk RNA-seq dataset were analyzed by ssGSEA for these senescence-related pathways (*Subramanian et al., 2005*). Finally, the significance of each senescence-related pathway in cancer and para-cancerous tissues was assessed using "wilcox.test".

The genes related to the G1/S phase (p15-Cell cycle G1/S, MDM2-p21-Cell cycle G1/S, and p27-Cell cycle G1/S; 27 genes in total) were extracted from the Kyoto Encyclopedia of Genes and Genomes (KEGG) official website. Subsequently, the scores for G1/S genes were calculated for each sample in TCGA dataset by "ssGSEA". Genes in the HALLMARK_G2M_CHECKPOINT pathway were downloaded from MSigDB *via* GSEA and the scores of G2 checkpoints were calculated for each sample in TCGA dataset by "ssGSEA". We downloaded REACTOME_TELOMERE_EXTENSION_BY_TELOMERASE using GSEA and the primary role of this pathway was "Telomere Extension By Telomerase". Finally, GSEA-derived enrichment scores in TCGA dataset for epithelial-mesenchymal transition (EMT) were computed by ssGSEA for 200 genes in the HALLMARK_EPITHELIAL_MESENCHYMAL_TRANSITION pathway for each sample (*Liberzon et al., 2015*).

## Univariate Cox and LASSO regression analyses

Univariate Cox analysis was conducted using the "coxph" function in the survival package of R to screen the significant ($p < 0.05$) prognosis-related genes. LASSO-Cox regression analysis was conducted for prognosis-related genes using the "glmnet" package in R (*Friedman, Hastie & Tibshirani, 2010*). The changing trajectory of each independent variable was analyzed. Increased lambda was positively related to increased number of

independent variable coefficients tending to zero. Finally, by 10-fold cross-validation, a model was constructed; for each lambda, the confidence intervals were calculated.

The risk-related prognostic score (Riskscore) was calculated for each sample according to the equation for the sample risk score: $Riskscore = \Sigma \beta i \times Expi$, where Expi denotes the level of gene expression of the corresponding gene signature and $\beta$ is the corresponding Cox regression coefficient. ROC analysis and z-score for RiskScore were computed using the "timeROC" package in R. Samples with RiskScore lower than zero comprised the low-risk group, while those with RiskScore greater than zero after z-score normalization were classified into the high-risk group. Finally, the Kaplan–Meier curves were plotted.

## Consensus clustering

Consensus clustering was conducted using the "ConsensusClusterPlus" package (*Wilkerson & Hayes, 2010*). In total, 500 bootstraps were conducted using the "hc" algorithm with "canberra" as the metric distance; in the training set, 80% of the patients were in each bootstrap process. After the number of clusters was selected to between 2 and 10, "ConsensusClusterPlus" in the TCGA clustered the 113 TNBC samples dataset. The optimal classification was confirmed based on the consensus matrix and cumulative distribution function (CDF).

## Mutation analysis

Using hg38 as the reference genome, the CNV-related TCGA-TNBC results were integrated by "gistic2″software at a confidence level of 0.9. The downloaded SNV data from TCGA were analyzed using the "maftools" (*Mayakonda et al., 2018*) package.

## Immune cell scoring and immunotherapy

The scores of the relevant immune cells were calculated using the "MCPcounter.estimate" function of the "MCPcounter" package (*Becht et al., 2016*), Tumor Immune Estimation Resource (TIMER) (*Li et al., 2020*) and Estimating the Proportion of Immune and Cancer cells (EPIC) (*Zhou et al., 2020*) methods. Significant differences were determined using "kruskal.test". Immune cell infiltration was assessed using the "Estimation of STromal and Immune cells in MAlignant Tumor tissues using Expression" (ESTIMATE) (*Yoshihara et al., 2013*) which provides information on tumor purity, scores of immune cell infiltration, and stromal cell levels in tumor tissues. Subsequently, the TIDE (*Jiang et al., 2018*) software was used to assess the putative immunotherapeutic efficacy in the defined molecular subtypes (http://tide.dfci.harvard.edu/). Higher TIDE score indicates less immunotherapy benefit as it suggests a greater possibility of immune escape. The correlation and significance of the risk score with the immune cell score were separately calculated using the "Hmisc" package's "rcorr" function based on Pearson's method.

## Differential expression analysis

Differential expression analysis of cluster 1, 2 and 3 was performed in both GSE and Target datasets in "limma" package (*Ritchie et al., 2015*). Finally, differentially expressed genes (DEGs) were filtered under "|log2 (Fold Change)| > 1 and $p < 0.05$".

## Quantitative reverse transcription-polymerase chain reaction

Using TRIzol reagent (Thermo Fisher, Waltham, MA, USA), total RNA was collected from MDA-MB-468, MDA-MB-231 and MCF10A cell lines with 260/280 the value was between 1.8–2.0. High-Capacity cDNA Reverse Transcription Kit (4368814, ThermoFisher, Waltham, MA, USA) was conducted for reverse transcription. Using a LightCycler 480 PCR System (Roche, Indianapolis, IN, USA), the RNA from each sample (2 µg) was subjected to qRT-PCR with FastStart Universal SYBR®Green Master (Roche, Indianapolis, IN, USA). The cDNA was a template with a reaction volume of 20 µl (0.5 µl of forward and reverse primers, 2 µl of cDNA template, 10 µl of PCR mixture, and appropriate volume of water). For the PCR reactions, cycling began with an initial DNA denaturation phase for 30 s (s) at 95 °C, then a total of 45 cycles were run for 15 s at 94 °C, for 30 s at 56 °C, and for 20 s at 72 °C. Each sample was performed for three separate analyses. Data from the threshold cycle (CT) were normalized to the level of GAPDH by $2^{-\Delta\Delta CT}$. The mRNA expression was compared in the normal and control tissues. Sequences of primer pairs targeting (Shanghai Gemma Gene Co., LTD, Shanghai, China) the genes are listed as follows:

| Gene | Forward primer sequence (5′-3′) | Reverse primer sequence (5′-3′) |
| --- | --- | --- |
| MMP28 | TCCCACCTCCACTCGATTCAG | GCCGCATAACTGTTGGTATCT |
| CT83 | CTCCTAGCGAGCAGCATTCTG | TTGATGACATTTCGCCAGTGT |
| ACP5 | TGAGGACGTATTCTCTGACCG | CACATTGGTCTGTGGGATCTTG |
| KRT6A | GAGGGTGAGCTACGTCCTTG | CAGCCGTATAGGTCTCTGTGT |
| GAPDH | AATGGGCAGCCGTTAGGAAA | GCCCAATACGACCAAATCAGAG |

## Cell culture and transient transfection

Breast cancer cell lines MDA-MB-468, MDA-MB-231, and non-tumorigenic epithelial cell lines MCF10A were suspended in serum-free cell freeze and immediately frozen in liquid nitrogen tanks. MDA-MB-468, MDA-MB-231 and MCF10A cell lines were commercially obtained from Beijing Bena Biotechnology Co. (Beijing, China). Cells were cultured in DEME F-12 medium (Gibco, Waltham, MA, USA). Lipofectamine 3000 (Invitrogen, Waltham, MA, USA) was used for transfecting the ACP5 siRNA (Sagon, China) and negative control (NC) into the cells. The target sequences for ACP5 siRNA were TCCTAAATCAAGCATCTTTCTGT (si ACP5).

## Transwell assay

Migration and invasion of MDA-MB-468 and MDA-MB-231cell lines were detected. Cell ($5 \times 10^4$) inoculation onto chambers coated (for invasion) or uncoated with Matrigel (BD Biosciences, USA) (for migration) was performed. The upper layer was added with serum-free medium, while the lower layer was added with complete DMEM medium. After incubation for 24 h, migrated or invaded cells were fixed using 4% paraformaldehyde and dyed by 0.1% crystalline violet. Cell number was counted using a light microscope. The experiment was conducted in our own lab.

## Statistical analysis

All statistical analysis was done with R program (v4.2.0). The Wilcoxon test evaluated two-group difference. Differences among three groups were analyzed by the Kruskal–Wallis test. Cox analysis and survival analysis both used the log-rank test. For Wet experiments, the number of each group was 3 independent experiments. Assay carried out by investigator's lab. Consumables in experiments we use are RNAase and DNAase free.

# RESULTS

## Nine annotated cell types displayed different senescence characteristics

A cellular senescence-related classifier was constructed for TNBC. The prognoses of patients with TNBC were verified using bulk RNA-seq and scRNA-seq data. The flow chart of the study design is presented in Fig. S1.

First, 38,582 cells were obtained by screening the data from nine samples (Table S1, Fig. 1A); these were clustered and 11 subpopulations were obtained (Fig. 1B). Subsequently, these cells were subjected to TSNE dimensionality reduction analysis, and the 11 subpopulations were annotated based on some classical markers of immune cells (Fig. 1C and Table S2). Among them, subpopulation C1 comprised CD8 T cells expressing CD8A, CD3D, GZMA, and CD8B; C6 comprised CD4 T cells expressing CD4 and CD3D; C2 and C9 comprised macrophages expressing CD163 and CD68; C3 and C10 comprised monocytes expressing S100A8; C5 comprised B cells expressing CD19, CD79A, and MS4A1; C4 comprised plasma cells expressing CD79A and JSRP1; C0 comprised epithelial cells expressing EPCAM; C3 and C8 contained fibroblasts expressing ACTA2, FAP, PDGFRB, and NOTCH3, and C7 contained endothelial cells expressing PECAM1. We also screened marker genes among these subpopulations and analyzed their corresponding expression (Fig. 1D). For instance, CD24 and KRT19 were highly expressed in epithelial cells in C0 and were the marker genes of C0.

The proportion of these nine subpopulations in each sample was further analyzed (Fig. 1E). Subsequently, changes in CNVs in single cells were predicted to distinguish normal from tumor cells in each sample (Fig. 1F). In total, 7,709 cancer cells and 30,298 normal cells were identified. Subsequently, the proportion of malignant and non-malignant cells in each sample was calculated (Fig. 1G). The cells in samples CID3946, CID44041, and CID4465 were all non-malignant, while the other samples comprised both malignant and non-malignant cells.

The senescence characteristics of single cells were further assessed. Cellular senescence-related pathway scores were higher in malignant relative to the non-malignant cells (Fig. 2).

## Validation of abnormal cellular senescence based on bulk RNA-seq data

Cellular senescence-related pathways were highly expressed in malignant cells at the single cell level relative to the non-malignant cells, with aberrant expression. Therefore, their expression in tumor and normal tissue samples was

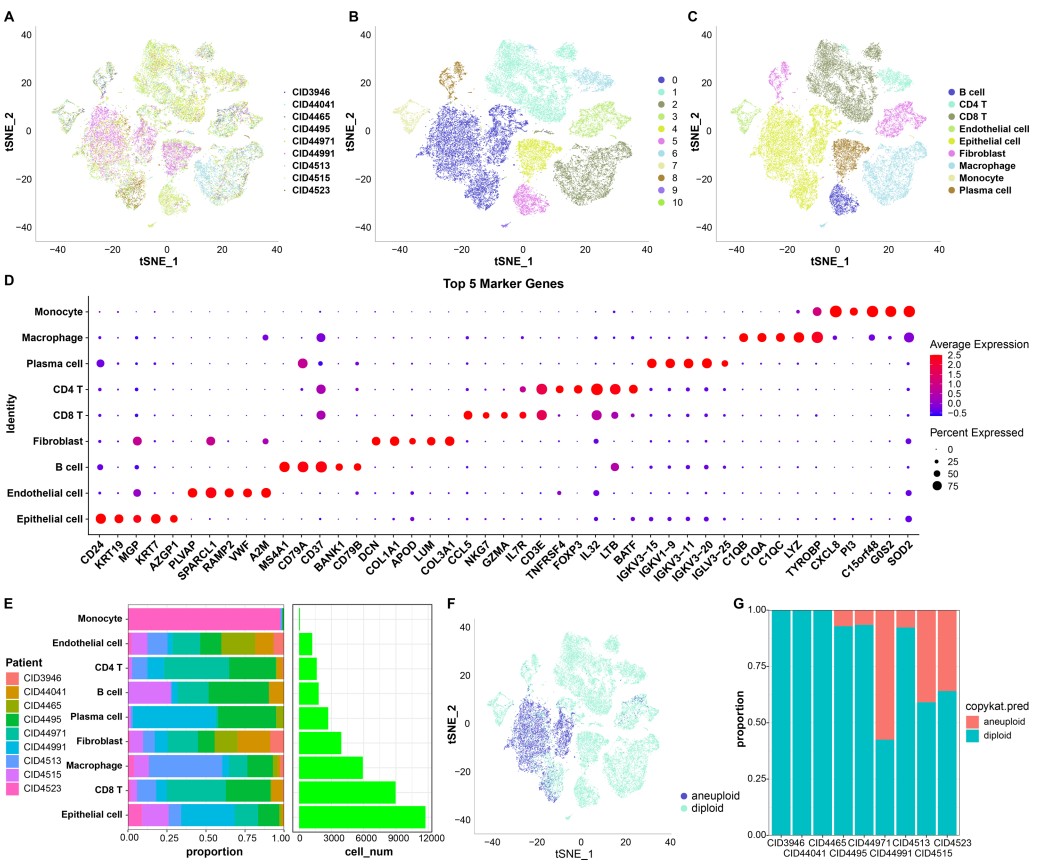

**Figure 1** **TSNE dimensionality-reduction results.** (A) TNSE plots for the distribution of nine samples; (B) TSNE plot for 11 immune cell subpopulations; (C) TSNE plot for cell distribution after annotation; (D) dot plots of the top five marker genes' expression in the annotated subpopulations; (E) proportions and cell count of the annotated subpopulations in each sample; (F) distribution of malignant and non-malignant cells predicted by "copykat"; (G) proportion of malignant and non-malignant cells in each sample.

further analyzed using a bulk RNA-seq dataset. GSEA results suggested that REACTOME_CELLULAR_SENESCENCE, and P53_SIGNALING_PATHWAY were significantly enriched in tumor tissues in TCGA dataset (Fig. 3A). ssGSEA of the bulk RNA-seq dataset (Fig. 3B) revealed that the enrichment scores of REACTOME_DNA_DAMAGE_TELOMERE_STRESS_INDUCED_SENESCENCE, P53_SIGNALING_PATHWAY, REACTOME_CELLULAR_SENESCENCE, REAC-TOME_DNA_DAMAGE_TELOMERE_STRESS_INDUCED_SENESCENCE were higher in some cancer tissues relative to the para-cancerous tissues. Therefore, these three cellular senescence-related pathways may exhibit regulatory complexity in cancerous tissues.

## Construction of cellular senescence-related subtypes

Three cellular senescence-related pathways, including REACTOME_CELLULAR_ SENESCENCE, and P53_SIGNALING_PATHWAY, REACTOME_DNA_DAMAGE_

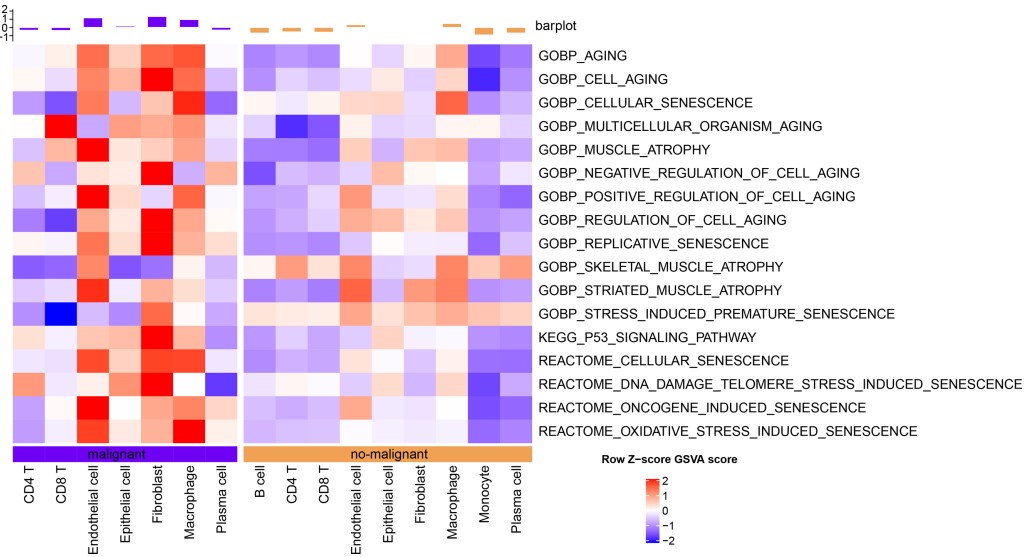

**Figure 2  Single-cell GSVA results of cellular senescence-related pathways in malignant and non-malignant cells.**

TELOMERE_STRESS_INDUCED_SENESCENCE, were significantly enriched in TNBC tissues. Therefore, we further analyzed the genes enriched in these three pathways.

The above pathways comprised 253 genes, of which 186 were present in TCGA dataset. Subsequently, a univariate analysis of these 186 genes yielded seven prognosis-related genes, including ETS2, SERPINE1, FOS, SHISA5, IL1A, TP53AIP1, and IGFBP7 (Fig. 4A). Mutations in these seven genes in TNBC were further examined. Among SNVs, *ETS2* showed the highest mutation frequency (primarily mutation type: missense) (Fig. 4B). Among CNVs, *IGFBP7* had the highest frequency of acquired CNVs (CNV_gain), whereas SHISA5 had the highest frequency of deletion mutations (CNV_loss) (Fig. 4C). The levels of these seven genes in normal *versus* tumor tissues were analyzed by the "wilcox.test" (Fig. 4D). The expressions of *SHISA5, SERPINE1*, and *IL1A* were markedly high in tumor tissues, whereas those of *TP53AIP1, ETS2*, and *FOS* were high in the normal tissues.

Consensus clustering of 113 TNBC samples in TCGA dataset was performed based on these seven key prognosis-related genes. Stable clustering results were achieved at $k = 3$ (Figs. 4E–4F). Therefore, three subtypes (clusters) were obtained (Fig. 4G). Significant differences in the prognostic characteristics among the subtypes were found (Fig. 4H). Overall, cluster 1 showed the best prognosis, followed by clusters 2 and 3. These seven genes showed the highest expression in cluster 3 (Fig. 4I).

## Differential analysis of cellular senescence-related subtypes

The expression of genes in these three senescence-related pathways among the three clusters was compared by "kruskal.test" (Fig. S2). Among the three pathways, genes in cluster 2 showed the lowest expression, while those in cluster 3 showed the highest expression.

The distribution of clinical characteristics across molecular subtypes was compared by chi-square test to assess the differences among the three subtypes in TCGA. The survival

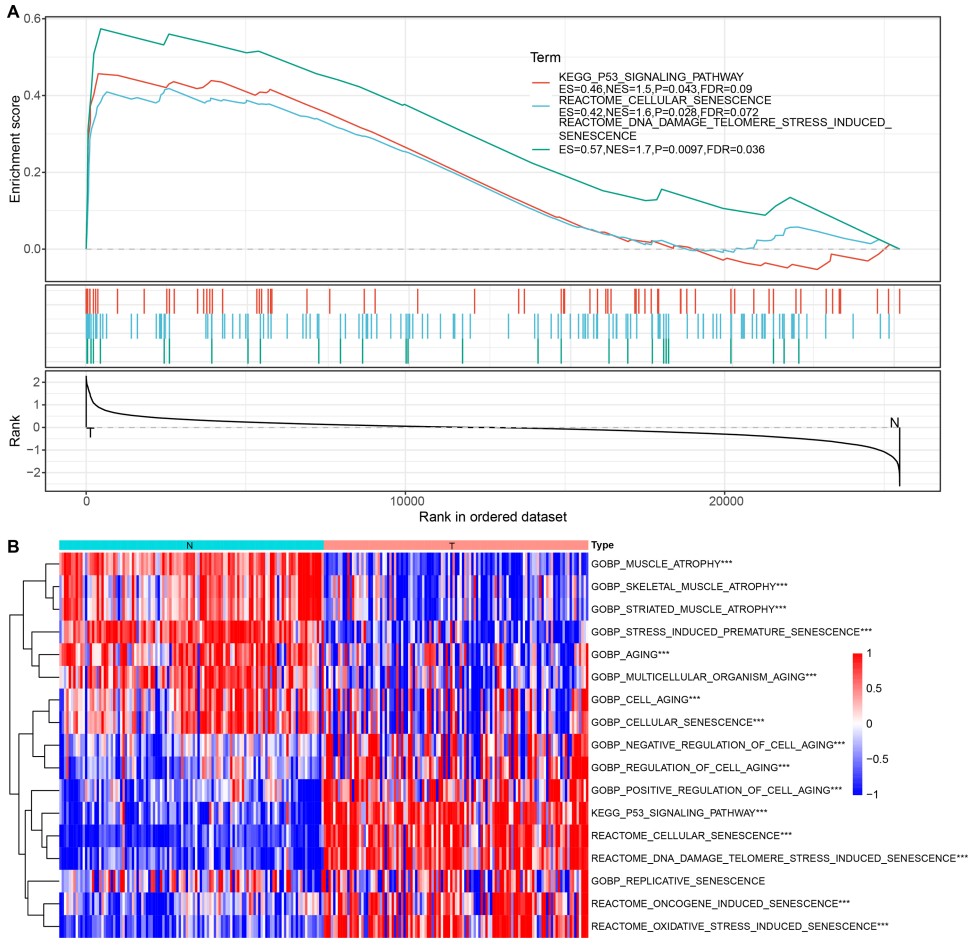

**Figure 3** **Enrichment analysis for bulk RNA-seq data.** (A) GSEA results in TCGA dataset; (B) Heat map of the distribution of cellular senescence-related pathway ssGSEA scores in cancerous and para-cancerous tissues in TCGA dataset.

statuses of these patients differed significantly among the three subtypes. In particular, the proportion of dead patients was high in cluster 3.

Differences in mutations among cellular senescence-related subtypes were analyzed. TCGA data suggested some differences in CNVs among the three subtypes (Figs. 5A and 5C). For instance, cluster 1 had the lowest degree of gene gain and lose. While cluster 3 exhibited the highest degree of CNV gain. Additionally, clusters 2 and 3 had similar CNV loss on chromosomes 2, 3, 4 and 5. Furthermore, *TP53, TTN, MUC16, SYNE1*, and *FAT3* had the highest SNV frequency (Fig. 5B).

## Biological characteristics of cellular senescence-related subtypes

Cancer cells can induce cellular senescence through the inhibition of the cell cycle, and cell cycle protein-dependent kinases (CDK inhibitory proteins), such as InK4a and p21, are upregulated in senescent cells, which causes cell cycle arrest (*Wang, Lankhorst & Bernards, 2022*). A previous study *Cuzick et al. (2011)* identified 31 genes associated with cell cycle

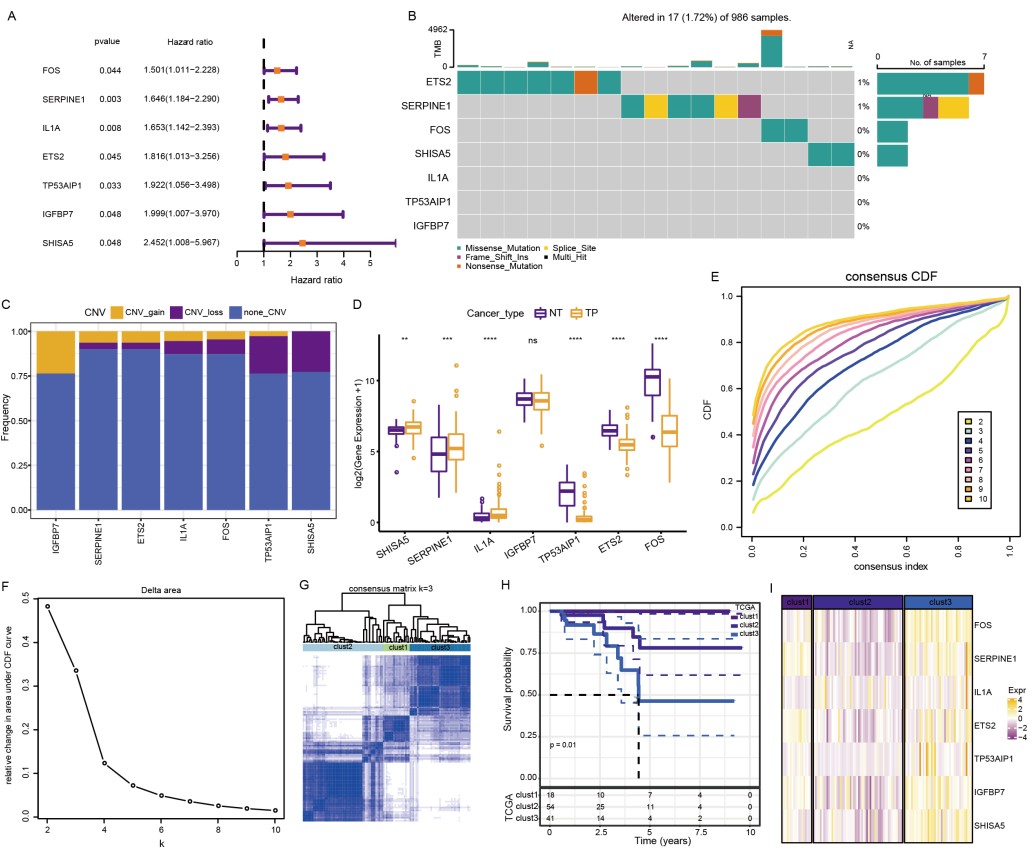

**Figure 4 Analysis of cellular senescence-related prognostic genes.** (A) Forest plot of prognosis-related genes; (B) waterfall plot of SNVs in the seven prognosis-related genes; (C) Percentage plot of CNVs in the seven prognosis-related genes; (D) The expression of the seven prognosis-related genes expressed in tumor and normal tissues; (E) CDF curve of TCGA cohort; (F) CDF-delta area curves for TCGA cohort. The Delta area curve for consensus clustering indicates the relative change in the area under the CDF curve for each category number, k, relative to k-1. The horizontal axis represents the category number, k, and the vertical axis represents the relative change in area under the CDF curve; (G) Heat map of sample clustering at consensus $k = 3$; (H) KM curve of the relationship between the prognoses of the three subtypes in TCGA; (I) Heat map of the expression of seven prognosis-related genes among three subtypes in TCGA dataset.

progression (CCP). We computed the CCP scores of each sample in TCGA dataset by ssGSEA. clusters 1 and 2 had slightly higher CCP scores relative to cluster 3 (wilcox.test) (Fig. 6A). No significant differences were observed among the three subtypes in the G1/S phase (wilcox.test) (Fig. 6B), whereas the scores for G2 checkpoints were slightly lower in cluster 3 relative to clusters 1 and 2 (wilcox.test) (Fig. 6C). Prognostic analysis suggested that patients in cluster 3 showed a worse prognosis. Therefore, the cell cycle may be one of the factors influencing cellular senescence. Other mechanisms may also regulate cellular senescence along with the cell cycle.

Similarly, inhibition of telomerase induces cellular senescence (*Wang, Lankhorst & Bernards, 2022*). Cancer cells usually circumvent telomere attrition by activating telomerase activity. Telomere elongation scores for telomerase in cluster 3 were lower than those in

Peer J

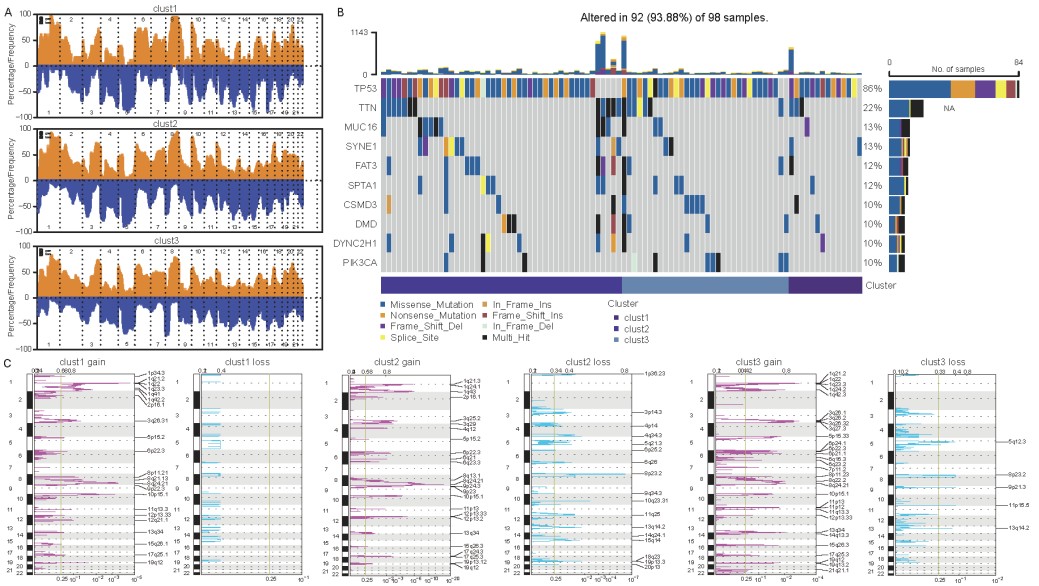

**Figure 5** **Differential analysis of cellular senescence-related subtype mutations.** (A) Comparison of CNVs among subtypes; (B) Waterfall plots of the top 10 genes with the highest SNVs among subtypes; (C) Peak plots of genes amplified (red) and missing (blue) among the three subtypes (G scores (top) and q-values (bottom) relative to the entire graph for amplification of all markers on the analyzed region).

clusters 1 and 2, with the latter showing the highest score. Therefore, other mechanisms may co-regulate cellular senescence with telomerase activity (wilcox.test) (Fig. 6D).

However, in addition to conveying the message "please kill me", the factors secreted by senescent cells also affect surrounding cells (*Yu et al., 2021*). This effect can promote tumor migration and metastasis by promoting EMT. Senescent tumor cells can promote the production of blood and lymphatic vessels by recruiting specialized macrophages that provide oxygen and nutrients necessary for growth to other tumor cells, thus promoting tumor growth and metastasis. We calculated the EMT scores. Cluster 2 had a lower EMT score than clusters 1 and 3, indicating that patients in cluster 3 were more likely to experience metastasis (wilcox.test) (Fig. 6E).

Based on genes in the HALLMARK_HYPOXIA pathway, the hypoxia scores of the samples were computed by ssGSEA. Angiogenesis scores of samples were also analyzed based on 24 genes from the literature (*Masiero et al., 2013*) (wilcox.test) (Figs. 6F–6G). Enrichment scores for 10 pathways associated with tumors in each sample in TCGA dataset were computed by the ssGSEA method described previously (*Sanchez-Vega et al., 2018*). Only two of the ten pathways (NOTCH and TGF-beta signaling) exhibited significant differences (wilcox.test).

Finally, genes in the HALLMARK_INFLAMMATORY_RESPONSE pathway from GSEA-based MSigDB were analyzed for inflammation-related scores by ssGSEA. Cluster 2 had a significantly lower inflammation-related score relative to clusters 1 and 3 (wilcox.test) (Fig. 6H).

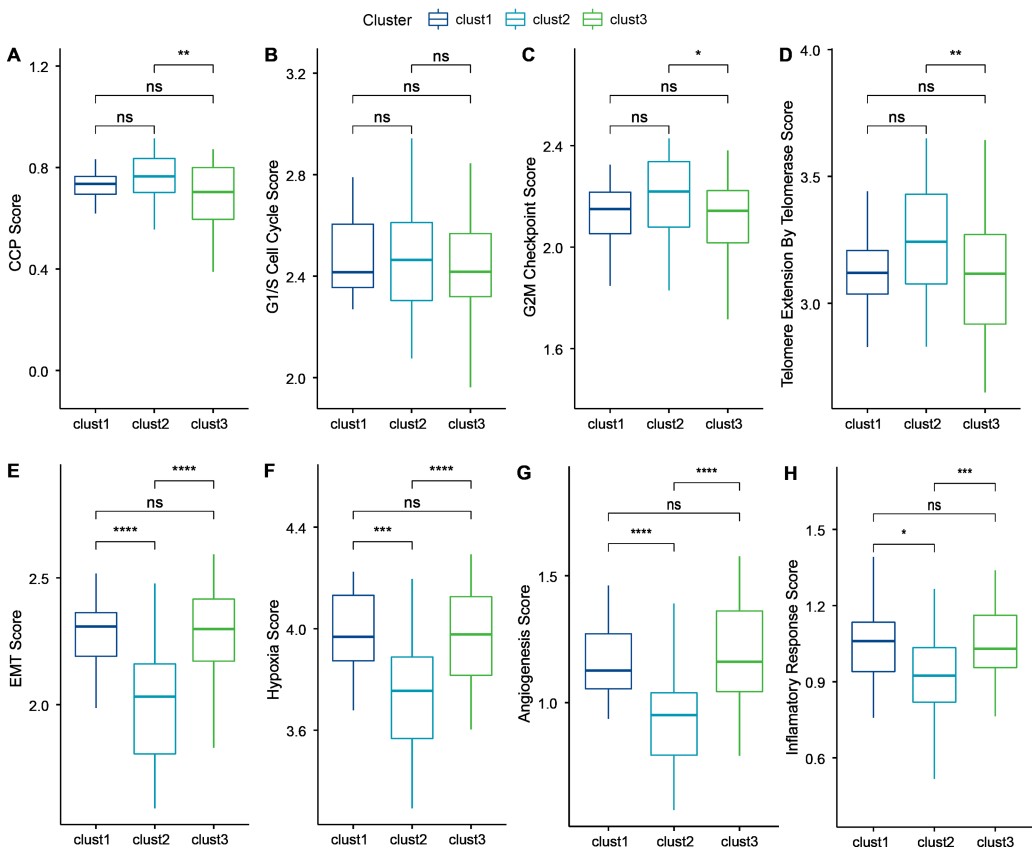

**Figure 6** Biological characterization of cellular senescence-related subtypes in TCGA-TNBC dataset. (A) CCP scores compared across three subtypes; (B) G1/S phase scores compared across three subtypes; (C) G2M-phase immune checkpoint scores compared across three subtypes; (D) Telomere extension scores of telomerase compared across three subtypes; (E) EMT scores among three different subtypes; (F) Hypoxic scores among three different subtypes; (G) Angiogenesis scores among three different subtypes; (H) Inflammatory factor scores among three different subtypes. ($p > 0.05$ denotes no significance; ****$p <$ 0.0001, ***$p < 0.001$, **$p < 0.01$, and *$p < 0.05$).

## Immune characteristics of cellular senescence-related subtypes

The relationship between cellular senescence-related subtypes and immunity was analyzed to investigate the immune characteristics of these subtypes. The immune scores of patients in TCGA dataset were calculated by "ESTIMATE". Significant differences were observed among the three subtypes, with a high degree of immune infiltration in clusters 1 and 3 (kruskal.test) (Fig. 7A).Subsequently, the scores for the relevant cells were calculated. Significant differences were noted for all cells except CD8 T cells and cytotoxic lymphocytes (kruskal.test) (Fig. 7B). Similarly, T cells CD4, also displayed distinct difference among three groups in TIMER and EPIC analysis (Figs. 7C–7D). Although the results were obtained from different algorithms, these results all together suggested higher immune infiltration status in clusters 3. The expression of immune checkpoint genes, including *NRP1, CD200, BTLA, LAIR1,* and *TNFRSF14* differed significantly among the three subtypes (kruskal.test) (Fig. 7E).

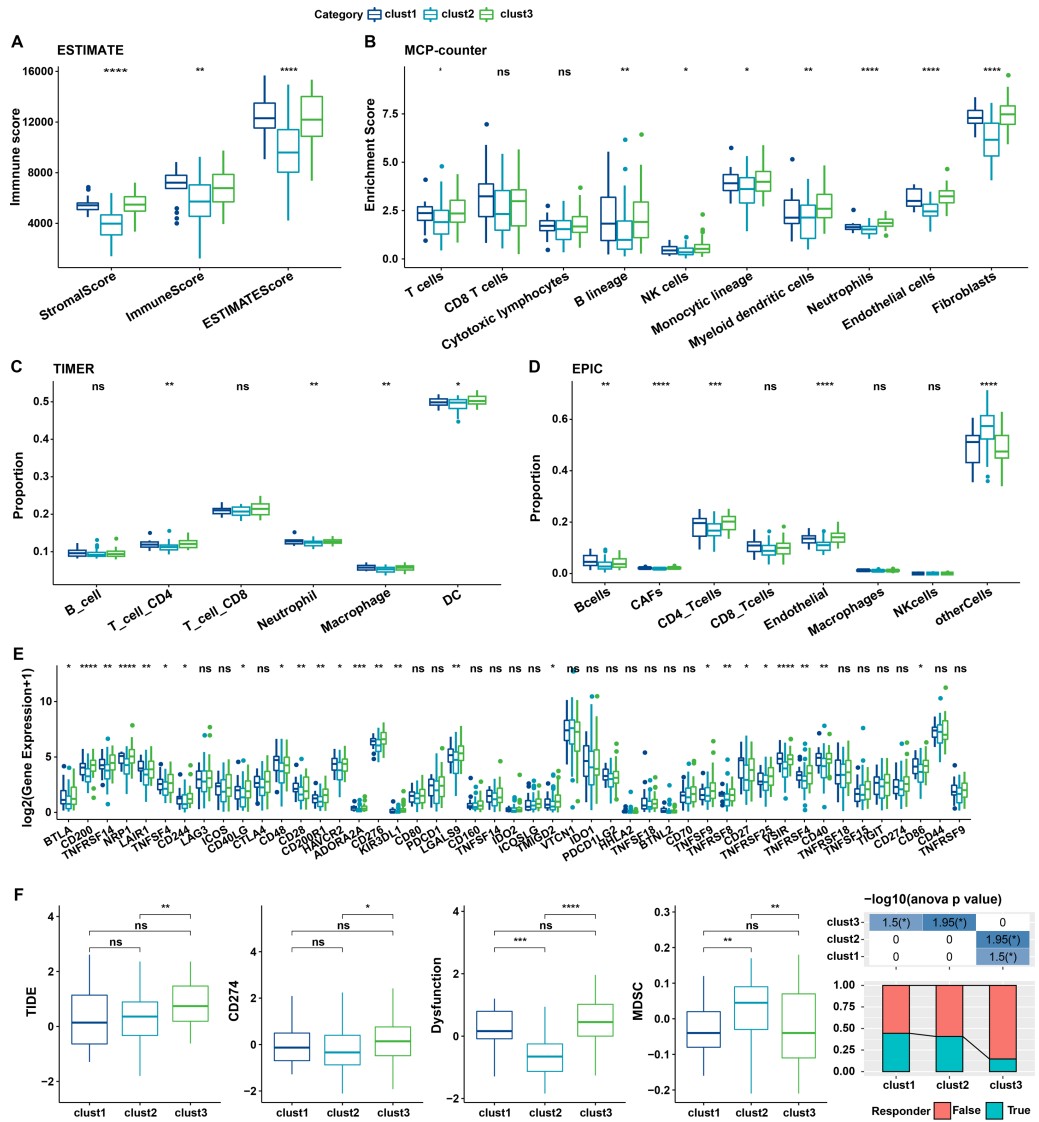

**Figure 7** **Immune characterization of cellular senescence-related subtypes in TCGA dataset.** (A) Comparison of ESTIMATE-predicted immune scores among the three subtypes (kruskal.test); (B) Comparison of MCPcounter-predicted cell scores among the three subtypes(kruskal.test); (C) Comparison of TIMER-predicted cell scores among the three subtypes (kruskal.test); (D) Comparison of EPIC-predicted cell scores among the three subtypes (kruskal.test); (E) Differences in immune checkpoint gene expression among the three subtypes (kruskal.test); (F) TIDE analysis for differential scores among the three subtypes (wilcox.test) ($p > 0.05$ denotes no significance; ****$p < 0.0001$, ***$p < 0.001$, **$p < 0.01$, and *$p < 0.05$).

Finally, the putative clinical efficacy of immunotherapy was assessed among these molecular subtypes. High TIDE scores in cluster 3 in TCGA cohort suggested that patients of this subtype were less likely to benefit from immunotherapy owing to a higher likelihood of immune escape (wilcox.test) (Fig. 7F).

## Identification of prognosis-related genes and construction of the risk model

Differential analysis was separately performed between cluster 1 and non-cluster 1; cluster 2 and non-cluster 2, and cluster 3 and non-cluster 3. Finally, 2 down-regulated and 34 up-regulated DEGs were screened between cluster 1 and non-cluster 1; 293 down-regulated and 9 up-regulated DEGs between cluster 2 and non-cluster 2, and 26 down-regulated and 218 up-regulated DEGs between cluster 3 and non-cluster 3. Thus, a total of 391 DEGs were screened.

Univariate Cox analysis of these 391 DEGs yielded 69 genes with high prognostic significance ($p < 0.05$), including 63 "Risk" and six "Protective" genes (Fig. S3A). These 69 hub genes were subsequently screened by LASSO regression (Fig. S3B). Further, using 10-fold cross-validation, a model was constructed and confidence intervals for each lambda were calculated (Fig. S3C). The model was optimum for lambda = 0.0782, whereby four genes were obtained, including *MMP28, CT83, ACP5,* and *KRT6A*; these were selected as the target genes for further analyses.

The four genes' expression levels were utilized to determine the risk score for each sample using the TCGA dataset as the training set. Then, using ROC analysis, the RiskScore prognostic classification was carried out (Fig. 8A). The prognostic prediction classification efficiency was analyzed separately for 1–5 years with obtained AUC values 0.91, 0.93, 0.74, 0.83 and 0.84, respectively , and the AUC values reached 0.7, indicating good predictive performance. Samples with RiskScore greater than zero after z-score normalization were classified into the high-risk group, whereas those with Riskscore lesser than zero comprised the low-risk group, with a highly significant ($p < 0.05$) difference between them (Fig. 8B).

To validate the model's robustness, the GSE58812 dataset was used for verification. Specifically, the risk model was constructed using these four genes and ROC analysis for prognostic classification was performed based on the RiskScore (Fig. 8C). The prognostic predictive classification efficiency was analyzed for 1–5 years with obtained AUC values 0.98, 0.71, 0.62, 0.66 and 0.65, respectively. All AUC values were over 0.6, displaying good predictive ability. Furthermore, a significant difference in prognosis was found between the risk groups ($p < 0.05$) (Fig. 8D).

## Validation of association between prognostic model signature and TNBC malignant phenotype

In this study, molecular assays revealed that the expression of *MMP28*, *ACP5* and *KRT6A* was increased in MDA-MB-468 and MDA-MB-231 cell lines. In contrast, the expression of CT83 was decreased in MDA-MB-468 and MDA-MB-231 cell lines (Figs. 9A–9D). This result is consistent with previous analyses that high expression of prognostic risk factors and low expression of protective factors associated with higher RiskScore. Given that *ACP5* is a key gene influencing high RiskScore, we inhibited the expression of *ACP5* in MDA-MB-468 and MDA-MB-231 cell lines, and by transwell assay, we could observe the ability of *ACP5* to promote the proliferation of triple-negative breast cancer cell lines. After we inhibited the expression of ACP5 in MDA-MB-468 and MDA-MB-231 cell lines, the invasion and migration ability of the cell lines were decreased (Figs. 9E–9G). This study not

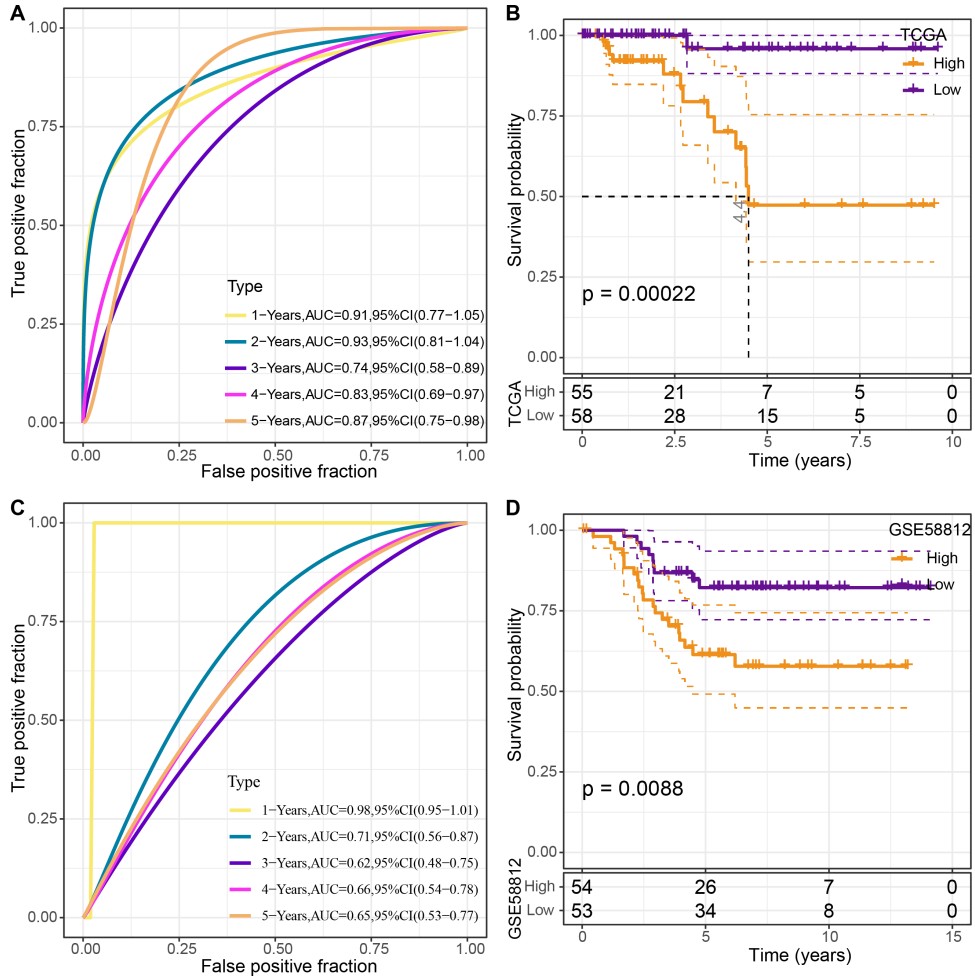

**Figure 8 Construction and validation of the risk model.** (A) ROC curves for the risk model constructed using four genes from TCGA dataset analysis; (B) KM curves for the risk model based on TCGA dataset; (C) ROC curves for the risk model constructed using four genes for the GSE58812 dataset; (D) KM curves for the risk model based on the GSE58812 dataset.

only confirms the existence of *ACP5* as a prognostic risk factor, but also demonstrates that this gene regulates the TNBC malignant phenotype at the cellular level, thereby influencing TNBC malignant progression.

## Immunological characteristics of RiskScore subgroups and sensitivity analysis based on drug IC50

To elucidate differences in the immune microenvironment of patients between the risk groups, differences in the relative cell abundances predicted by "MCPcounter" were compared (wilcox.test) (Fig. 10A). T cells, cytotoxic lymphocytes, and monocytic lineage differed significantly between the risk groups. Immune cell infiltration was also assessed (wilcox.test) (Fig. 10B). "ImmuneScore" was higher in the high-risk group relative to the low-risk group, suggesting that patients in the former had high immune cell infiltration.

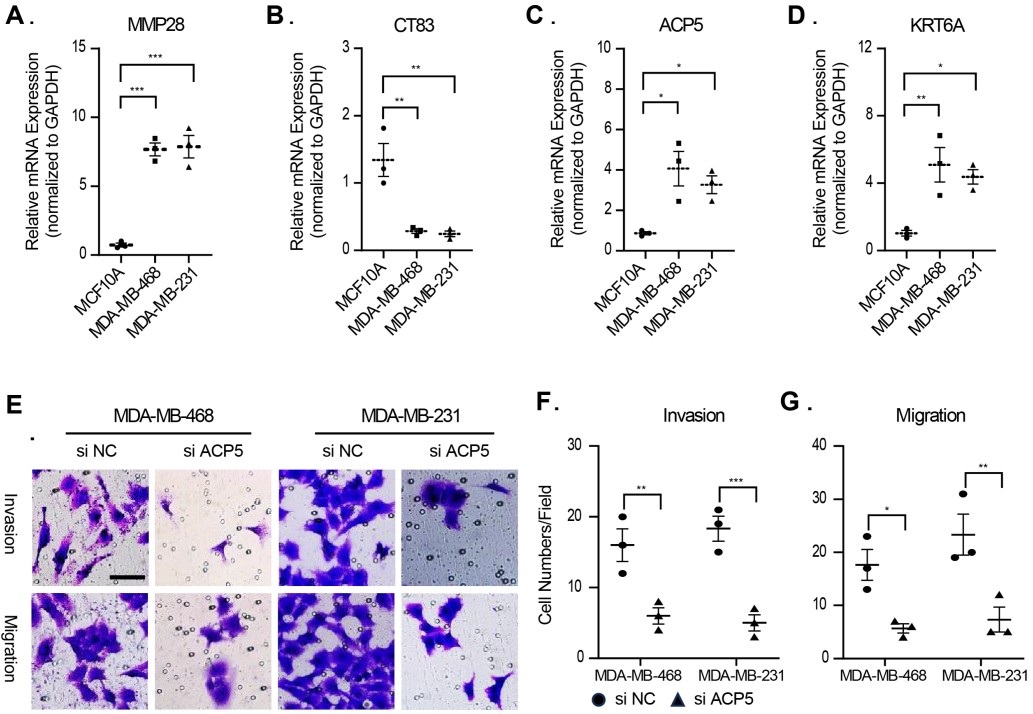

**Figure 9** **The validation of cellular regulation function of signature.** (A–D) The mRNA expression levels of MMP28, CT83, ACP5 and KRT6A in MCF-10A, MDA-MB-468 and MDA-MB-231 cell lines. (E) Representative migration and invasion images of MDA-MB-468 and MDA-MB-231 cell lines after inhibition of ACP5. (F–G) Quantitative analysis for migration and invasion ability. $n = 3$, *$\leq$0.05, **$\leq$0.01, ***$\leq$0.001. The results are presented as mean ± S.E.M.

Additionally, TIMER and EPIC analysis also revealed higher immune infiltration status in high risk group (Figs. 10C–10D).

Subsequently, differences in response to immunotherapy between the risk groups were analyzed in TCGA cohort. First, differences in immune checkpoint expression were compared. Most immune checkpoint genes, including *TNFRSF14, LAIR1,* and *CD244* showed differential expression between the risk groups, with high expression in the high-risk group (wilcox.test) (Fig. 10E).

A significant positive correlation between the Riskscore and T cells, cytotoxic lymphocytes, and monocytic lineage (Fig. 10F) was observed. Risk scores also correlated positively with "ImmuneScore", "StromalScore", and "ESTIMATEScore" (Fig. 10G). Drug sensitivity prediction revealed that low-risk group showed higher sensitivity to six traditional drugs (MG-132, Sorafenib, A77004, Bortezomib, Shikonin and AZ628) by wilcox.test ($p < 0.05$, Fig. 10H).

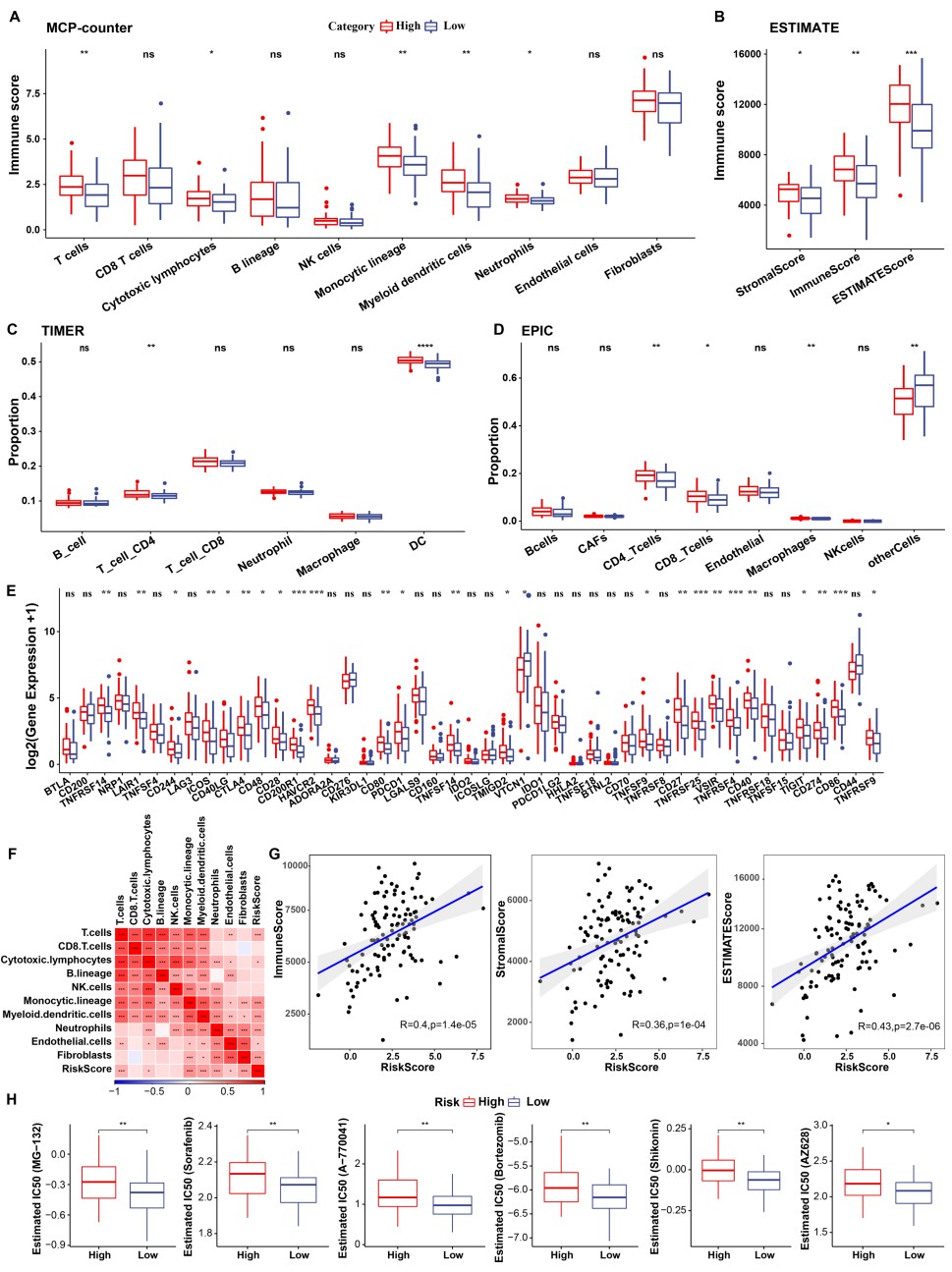

**Figure 10   Immunological characteristics of RiskScore subgroups and sensitivity analysis in TCGA-TNBC cohort.** (A) Differences in MCPcounter-predicted cell scores between risk groups; (B) Differences in immune and stromal scores between risk groups; (C) comparison of TIMER-predicted cell scores between risk groups; (D) comparison of EPIC-predicted cell scores between risk groups; (E) Differentially expressed immune checkpoints between risk groups; (F) Correlation analysis of cell scores with risk scores; (G) Correlation analysis of ImmuneScore, StromalScore, and ESTIMATEScore with risk scores; (H) The box plot of estimated IC50 for drugs (wilcox.test; $p > 0.05$ denotes no significance, $^{*}p < 0.05$, $^{**}p < 0.01$, $^{***}p < 0.001$, and $^{****}p < 0.0001$).

Peer J

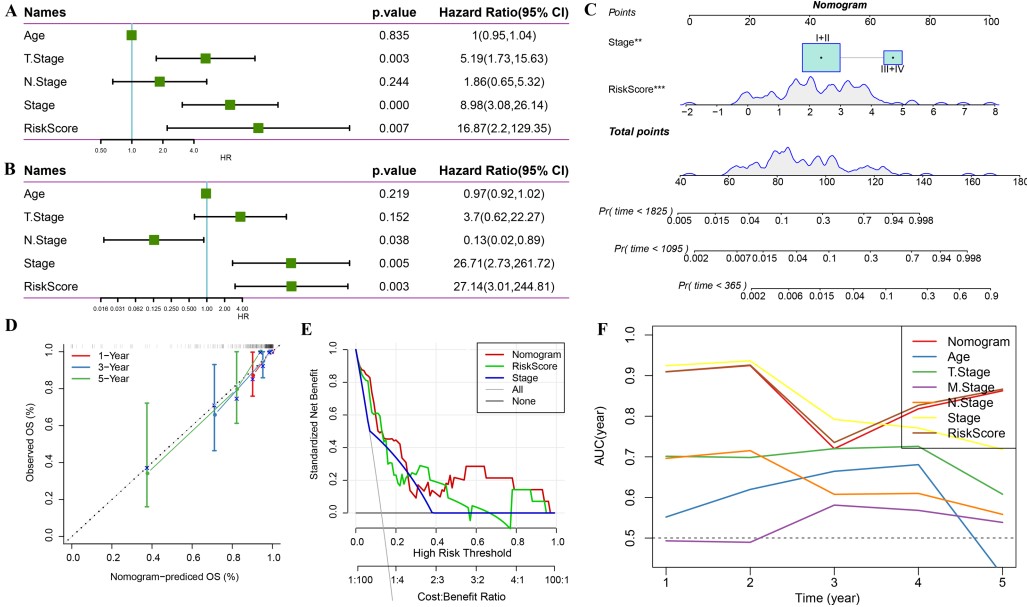

**Figure 11 Prognostic model and survival prediction.** (A–B) Univariate and multivariate Cox analyses of RiskScore and clinicopathological characteristics; (C) Nomogram model; (D) 1-, 3-, and 5-year calibration curves for the nomogram; (E) Decision curve for the nomogram; (F) Relative to other clinicopathological features, the nomogram exhibits the best capacity for survival prediction.

## Combination of RiskScore and clinicopathological characteristics improves the performance of the prognostic model and its survival predictive accuracy

Univariate regression analysis based on RiskScore and other clinicopathological characteristics suggested that the latter was the most significant prognostic factor (*p*-value = 0.007, Fig. 11A) and remained so (*p*-value = 0.003, Fig. 11B). A nomogram was constructed to quantify the survival probability of patients and risk assessment by combining RiskScore and other clinicopathological characteristics (Fig. 11C). The results confirmed the greatest impact of RiskScore on survival prediction. Subsequently, the model's prediction accuracy was assessed using a calibration curve (Fig. 11D). The findings indicated that the calibration curves for 1-, 3-, and 5-year predictions nearly overlapped with the standard curves, pointing to the nomogram's strong predictive ability. Furthermore, decision curve analysis (DCA) was used to evaluate the model's dependability. DCA results highlighted that both RiskScore and nomogram had significantly more benefits than the extreme curves with strong survival predictive power relative to other clinicopathological features, especially for the 5-year survival (Figs. 11E–11F).

## DISCUSSION

TNBC is a highly heterogeneous type of cancer (*Bianchini et al., 2016*). Recent studies have classified TNBC based on different molecular or clinical subtypes and identified possible therapeutic targets, primarily based on transcriptomic subtypes (*Metzger-Filho et al.,*

*2012*; *Garrido-Castro, Lin & Polyak, 2019*; *Hu et al., 2022*). Cellular senescence promotes tumor regression through cell-non-autonomous and -autonomous mechanisms. Drugs inducing cancer cell senescence and modulating SASPs provide new targets for tumor therapy (*Calcinotto et al., 2019*). However, no cellular senescence-based typing of TNBC has been established to date. In this study, we proposed cellular senescence molecular typing for the first time, and identified seven genes related to TNBC prognosis through analyses and established three molecular subtypes. Subsequently, DEGs among three molecular subtypes was adopted for the construction of risk model. Finally, a nonogram for clinical decision-making was designed.

Seven cellular senescence related genes exhibited different biological functions in TNBC or other cancer development. *ETS2* is a key ETS transcriptional family member implicated in invasion and metastasis in TNBC (*Kollareddy & Martinez, 2021*). A screening experiment conducted against *CDK10* confirmed the *ETS2* transcription factor as an interacting protein (*Guen et al., 2017*). This finding indicated that ETS2 may also participate in cell cycle process. Serpin family E member 1 (*SERPINE1*) is responsible for encoding plasminogen activator inhibitor 1, which is a primary inhibitor of tissue plasminogen activator (*Li et al., 2018*). SERPINE1 has been detected in various cancer and involved in cancer invasion, migration, and angiogenesis (*Seker et al., 2019*; *Yang, Ma & Zhu, 2019*). Also, *SERPINE1*, a paclitaxel-resistant oncogene, is a putative target for TNBC treatment (*Zhang, Lei & Jing, 2020b*). *FOS*, a key regulator of TNBC cell proliferation and viability, is related to a poor patient prognosis (*Zhang, Lei & Jing, 2020a*). *IL1A*, a pro-inflammatory cytokine, enhances cell growth and invasiveness of TNBC cells. Abnormal IL1A induction correlates with a poor patient prognosis in TNBC (*You et al., 2021*). *IGFBP7* was reported to suppress breast tumor growth through the induction of apoptosis and senescence pathways (*Benatar et al., 2012*). Overall, these findings disclosed the potential functions or the prognostic ability of selected hub genes and supported the reliability of our results.

TNBC is the most aggressive and heterogeneous subtype of breast cancer, and consists of several different microenvironmental phenotypes (*Xiao et al., 2019*; *Gruosso et al., 2019*; *Bareche et al., 2020*). Previous studies have classified several molecular subtypes for TNBC. For instance, Lehmann and co-workers carried out clustering analysis and categorized TNBC patients into six subtypes, called basal-like 1, basal-like 2, immunomodulatory, luminal androgen receptor, mesenchymal, and mesenchymal stem-like subclusters (*Lehmann et al., 2011*). Next, a transcriptome-wide analysis of Chinese TNBC samples clustered TNBC patients into four subtypes - BLIS, IM, LAR, and MES (*Tong et al., 2023*). In this research, we classified TNBC patients into only three clusters based on senescence related characteristics, and our immunological analysis highlighted significant differences in TME among the three subtypes, which could support TNBC cellular senescence-based therapy and support clinical decision-making. Changes in the vital immune cells that make up the stroma around the tumor are correlated with tumor growth. TME metabolism changes are the result of the aberrant metabolic status of tumor cells (*Hinshaw & Shevde, 2019*). The immune score produced by the ESTIMATE algorithm rates the immunological make-up and speed of tumor sample responses. Strong correlations exist between prognosis and tumor purity, or the proportion of malignant cells
in the tumor tissue (*Aran, Sirota & Butte, 2015*). Here, we used "ESTIMATE" to identify three subtypes in TME. Significant differences were found among the three subtypes, with patients in clusters 1 and 3 showing a high degree of immune infiltration. Actually, senescent cells secrete various factors which make uo senescence-relevant secretory phenotype, including pro-inflammatory factors (cytokines, chemokines, micro-RNAs) to induce inflammatory state (*Frasca et al., 2021*).Therefore, in this work, TME-related cellular senescence characteristics are of interest in determining treatment strategies for TNBC and could be potential targets for individualized therapy.

We built a risk model based on genes in cellular senescence-related pathways and calculated risk scores based on four significant genes, including *MMP28, CT83, ACP5*, and *KRT6A*, and validated the relative expression levels of the four genes in TNBC cells. A previous study defined a subset of TNBC with poor prognosis based on the basal marker, *KRT16*, the stem cell marker, *WNT11*, and the EMT marker, *MMP28* (*Yu et al., 2013*), corroborating the rationality and validity of our gene selection approach. We constructed a risk model incorporating multiple genes, wherein *MMP28* was a risk factor. EMT scores were calculated. Patients in cluster 2 had lower EMT scores than those in clusters 1 and 3, suggesting that the former were more likely to experience metastasis. *CT83* is a gene specific to TNBC and its hypermethylation is oncogenic in breast cancer (*Chen et al., 2021*). Overexpression of four genes, including *ART3, FABP7, TTYH1,* and *CT83* correlated positively with patients' life expectancy rates in TNBC ($p < 0.05$) (*Zhong et al., 2020*). This is supported by our study revealing the down-regulated expression of this gene in the high-risk group. A model comprising *CT83* and *FABP7* constructed using Cox regression, Kaplan–Meier, and ROC analyses correlated significantly with OS in TNBC. *ACP5* has been shown to correlate with poor patient prognosis in cancers such as gastric and rectal cancers, and to correlate with malignant tumor progression by affecting malignant phenotypes such as cell proliferation and invasion (*Bian et al., 2019*; *An et al., 2021*). In breast cancer-related studies, the *TRAP/ACP5/*uteroferrin/purple acid phosphatase/PP5 signaling axis can act as a driver mediating breast cancer invasion and mediate cancer malignant progression at the cellular level (*Krumpel et al., 2015*). This informs the present study, in which we reveal that this gene is up-regulated for expression in TNBC cells and promotes cell migration and invasive ability, providing a reference for subsequent work to uncover the mechanisms by which this gene affects TNBC progression at the cellular level. Ultimately, we identified potential biomarkers implicated in the onset and progression of TNBC. These genes may provide novel insights into the tumorigenesis of TNBC and serve as independent prognostic factors for TNBC. Taken together, these findings demonstrate the validity of the risk model developed in this study.

Some limitations exist in this study that should be stated prior to outlining the conclusions. Firstly, we could not demonstrate the difference and role of these subtypes in TNBC progression because of a lack of progression data such as tumor stage of the patients. Second, we only verified the role of the key signature gene associated with RiskScore in the regulation of the malignant phenotype of TNBC cells at the cellular level, but we did not verify the molecular mechanisms of the signature gene, such as influencing cytokine secretion to regulate cellular senescence, to corroborate the results of the bioinformatic

analyses. Finally, rather than evaluating our cohort, the data were taken from available databases. As a result, there was still little evidentiary impact. Prospective studies are thus needed to validate the therapeutic and prognostic value of the established subtypes in TNBC.

## CONCLUSION

In summary, we elucidated the abnormalities among cellular senescence-related pathways in TME of malignant cells of TNBC. Senescence-related subtypes and a risk model were constructed based on the genes in these pathways. Also, we validated the oncogenic gene *ACP5*'s ability to migrate and invade in TNBC cells. Additionally, a nomogram was designed for future clinical decision-making. All these findings provide a basis for the revelation of the molecular mechanisms by which senescence-associated genes influence TNBC progression and directions for TNBC therapy and prognosis.

### Abbreviations

| | |
|---|---|
| **CDF** | Cumulative distribution function |
| **CNV** | Copy number variation |
| **DCA** | Decision curve analysis |
| **ESTIMATE** | Estimation of STromal and Immune cells in MAlignant Tumor tissues using Expression data |
| **GEO** | Gene Expression Omnibus |
| **KEGG** | Kyoto Encyclopedia of Genes and Genomes |
| **Lasso** | Least absolute shrinkage and selection operator |
| **SASP** | Senescence-associated secretory phenotype |
| **scRNA-seq** | single-cell RNA sequencing |
| **SNV** | Single nucleotide variant |
| **ssGSEA** | single sample gene set enrichment analysis |
| **TCGA** | The Cancer Genome Atlas |
| **TNBC** | Triple-negative breast cancer |

### Funding

The authors received no specific funding for this work.

### Competing Interests

The authors declare there are no competing interests.

### Author Contributions

- Tengfei Cao conceived and designed the experiments, analyzed the data, prepared figures and/or tables, authored or reviewed drafts of the article, and approved the final draft.
- Mengjie Huang conceived and designed the experiments, performed the experiments, authored or reviewed drafts of the article, and approved the final draft.

- Xinyue Huang performed the experiments, analyzed the data, prepared figures and/or tables, and approved the final draft.
- Tian Tang analyzed the data, authored or reviewed drafts of the article, and approved the final draft.

## Data Availability

The datasets are available at GEO: GSE58812, GSE176078.

The raw data is available at GitHub and Zenodo:

- https://github.com/ctf1985/Raw-and-Experimental-Data.git, 10.5281/zenodo.10146272

- ctf1985. (2023). ctf1985/Raw-and-Experimental-Data: First release of my source data (v.1.0.0). Zenodo. https://doi.org/10.5281/zenodo.10146272.

## Supplemental Information

Supplemental information for this article can be found online at http://dx.doi.org/10.7717/peerj.16935#supplemental-information.

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

# PeerJ

**Bavik C, Coleman I, Dean JP, Knudsen B, Plymate S, Nelson PS. 2006.** The gene expression program of prostate fibroblast senescence modulates neoplastic epithelial cell proliferation through paracrine mechanisms. *Cancer Research* **66**(2):794–802 DOI 10.1158/0008-5472.CAN-05-1716.

**Becht E, Giraldo NA, Lacroix L, Buttard B, Elarouci N, Petitprez F, Selves J, Laurent-Puig P, Sautès-Fridman C, Fridman WH, De Reyniès A. 2016.** Estimating the population abundance of tissue-infiltrating immune and stromal cell populations using gene expression. *Genome Biology* **17**(1):218 DOI 10.1186/s13059-016-1070-5.

**Benatar T, Yang W, Amemiya Y, Evdokimova V, Kahn H, Holloway C, Seth A. 2012.** IGFBP7 reduces breast tumor growth by induction of senescence and apoptosis pathways. *Breast Cancer Research and Treatment* **133**(2):563–573 DOI 10.1007/s10549-011-1816-4.

**Bhatia B, Multani AS, Patrawala L, Chen X, Calhoun-Davis T, Zhou J, Schroeder L, Schneider-Broussard R, Shen J, Pathak S, Chang S, Tang DG. 2008.** Evidence that senescent human prostate epithelial cells enhance tumorigenicity: cell fusion as a potential mechanism and inhibition by p16INK4a and hTERT. *International Journal of Cancer* **122**(7):1483–1495 DOI 10.1002/ijc.23222.

**Bian ZQ, Luo Y, Guo F, Huang YZ, Zhong M, Cao H. 2019.** Overexpressed ACP5 has prognostic value in colorectal cancer and promotes cell proliferation and tumorigenesis via FAK/PI3K/AKT signaling pathway. *American Journal of Cancer Research* **9**(1):22–35.

**Bianchini G, Balko JM, Mayer IA, Sanders ME, Gianni L. 2016.** Triple-negative breast cancer: challenges and opportunities of a heterogeneous disease. *Nature Reviews Clinical Oncology* **13**(11):674–690 DOI 10.1038/nrclinonc.2016.66.

**Borodkina AV, Shatrova AN, Nikolsky NN, Burova EB. 2016.** Role of P38 map-kinase in the stress-induced senescence progression of human endometrium-derived mesenchymal stem cells. *Tsitologiia* **58**(6):429–435.

**Calcinotto A, Kohli J, Zagato E, Pellegrini L, Demaria M, Alimonti A. 2019.** Cellular senescence: aging, cancer, and injury. *Physiological Reviews* **99**(2):1047–1078 DOI 10.1152/physrev.00020.2018.

**Campisi J, Andersen JK, Kapahi P, Melov S. 2011.** Cellular senescence: a link between cancer and age-related degenerative disease? *Seminars in Cancer Biology* **21**(6):354–359.

**Castro-Vega LJ, Jouravleva K, Ortiz-Montero P, Liu WY, Galeano JL, Romero M, Popova T, Bacchetti S, Vernot JP, Londoño Vallejo A. 2015.** The senescent microenvironment promotes the emergence of heterogeneous cancer stem-like cells. *Carcinogenesis* **36**(10):1180–1192 DOI 10.1093/carcin/bgv101.

**Chakrabarty A, Chakraborty S, Bhattacharya R, Chowdhury G. 2021.** Senescence-induced chemoresistance in triple negative breast cancer and evolution-based treatment strategies. *Frontiers in Oncology* **11**:674354 DOI 10.3389/fonc.2021.674354.

**Chen C, Gao D, Huo J, Qu R, Guo Y, Hu X, Luo L. 2021.** Multiomics analysis reveals CT83 is the most specific gene for triple negative breast cancer and its

hypomethylation is oncogenic in breast cancer. *Scientific Reports* **11(1)**:12172
DOI 10.1038/s41598-021-91290-4.

**Collado M, Blasco MA, Serrano M. 2007.** Cellular senescence in cancer and aging. *Cell*
**130(2)**:223–233 DOI 10.1016/j.cell.2007.07.003.

**Coppé JP, Desprez PY, Krtolica A, Campisi J. 2010.** The senescence-associated secre-
tory phenotype: the dark side of tumor suppression. *Annual Review of Pathology*
**5**:99–118 DOI 10.1146/annurev-pathol-121808-102144.

**Coppé JP, Patil CK, Rodier F, Sun Y, Muñoz DP, Goldstein J, Nelson PS, Desprez
PY, Campisi J. 2008.** Senescence-associated secretory phenotypes reveal cell-
nonautonomous functions of oncogenic RAS and the p53 tumor suppressor. *PLOS
Biologyogy* **6(12)**:2853–2868.

**Cuzick J, Swanson GP, Fisher G, Brothman AR, Berney DM, Reid JE, Mesher D,
Speights VO, Stankiewicz E, Foster CS, Møller H, Scardino P, Warren JD, Park
J, Younus A, Flake 2nd DD, Wagner S, Gutin A, Lanchbury JS, Stone S. 2011.**
Prognostic value of an RNA expression signature derived from cell cycle proliferation
genes in patients with prostate cancer: a retrospective study. *The Lancet Oncology*
**12(3)**:245–255 DOI 10.1016/S1470-2045(10)70295-3.

**Dagogo-Jack I, Shaw AT. 2018.** Tumour heterogeneity and resistance to cancer therapies.
*Nature Reviews Clinical Oncology* **15(2)**:81–94 DOI 10.1038/nrclinonc.2017.166.

**Foster SA, Wong DJ, Barrett MT, Galloway DA. 1998.** Inactivation of p16 in human
mammary epithelial cells by CpG island methylation. *Molecular and Cellular Biology*
**18(4)**:1793–1801 DOI 10.1128/MCB.18.4.1793.

**Foulkes WD, Smith IE, Reis-Filho JS. 2010.** Triple-negative breast cancer. *The New
England Journal of Medicine* **363(20)**:1938–1948 DOI 10.1056/NEJMra1001389.

**Frasca D, Saada YB, Garcia D, Friguet B. 2021.** Effects of cellular senescence on
metabolic pathways in non-immune and immune cells. *Mechanisms of Ageing and
Development* **194**:111428 DOI 10.1016/j.mad.2020.111428.

**Friedman J, Hastie T, Tibshirani R. 2010.** Regularization paths for generalized linear
models via coordinate descent. *Journal of Statistical Software* **33(1)**:1–22.

**Gao R, Bai S, Henderson YC, Lin Y, Schalck A, Yan Y, Kumar T, Hu M, Sei E,
Davis A, Wang F, Shaitelman SF, Wang JR, Chen K, Moulder S, Lai SY, Navin
NE. 2021.** Delineating copy number and clonal substructure in human tu-
mors from single-cell transcriptomes. *Nature Biotechnology* **39(5)**:599–608
DOI 10.1038/s41587-020-00795-2.

**Garrido-Castro AC, Lin NU, Polyak K. 2019.** Insights into molecular classifications
of triple-negative breast cancer: improving patient selection for treatment. *Cancer
Discovery* **9(2)**:176–198 DOI 10.1158/2159-8290.CD-18-1177.

**Gluz O, Liedtke C, Gottschalk N, Pusztai L, Nitz U, Harbeck N. 2009.** Triple-
negative breast cancer–current status and future directions. *Annals of Oncology*
**20(12)**:1913–1927 DOI 10.1093/annonc/mdp492.

**Gribov A, Sill M, Lück S, Rücker F, Döhner K, Bullinger L, Benner A, Unwin A. 2010.**
SEURAT: visual analytics for the integrated analysis of microarray data. *BMC
Medical Genomics* **3**:21 DOI 10.1186/1755-8794-3-21.

**Gruosso T, Gigoux M, Manem VSK, Bertos N, Zuo D, Perlitch I, Saleh SMI, Zhao H, Souleimanova M, Johnson RM, Monette A, Ramos VM, Hallett MT, Stagg J, Lapointe R, Omeroglu A, Meterissian S, Buisseret L, Eynden GVanden, Salgado R, Guiot MC, Haibe-Kains B, Park M. 2019.** Spatially distinct tumor immune microenvironments stratify triple-negative breast cancers. *Journal of Clinical Investigation* **129(4)**:1785–1800 DOI 10.1172/JCI96313.

**Guen VJ, Gamble C, Lees JA, Colas P. 2017.** The awakening of the CDK10/Cyclin M protein kinase. *Oncotarget* **8(30)**:50174–50186 DOI 10.18632/oncotarget.15024.

**Han Z, Wang Y, Han L, Yang C. 2023.** RPN2 in cancer: an overview. *Gene* **857**:147168 DOI 10.1016/j.gene.2023.147168.

**Hinshaw DC, Shevde LA. 2019.** The tumor microenvironment innately modulates cancer progression. *Cancer Research* **79(18)**:4557–4566.

**Hollstein M, Sidransky D, Vogelstein B, Harris CC. 1991.** p53 mutations in human cancers. *Science* **253(5015)**:49–53 DOI 10.1126/science.1905840.

**Hu L, Chen M, Dai H, Wang H, Yang W. 2022.** A metabolism-related gene signature predicts the prognosis of breast cancer patients: combined analysis of high-throughput sequencing and gene chip data sets. *Oncologie* **24(4)**:803–822 DOI 10.32604/oncologie.2022.026419.

**Huang L, Jian Z, Gao Y, Zhou P, Zhang G, Jiang B, Lv Y. 2019.** RPN2 promotes metastasis of hepatocellular carcinoma cell and inhibits autophagy via STAT3 and NF-kappaB pathways. *Aging* **11(17)**:6674–6690 DOI 10.18632/aging.102167.

**Huang X, Wang X, Qian H, Jin X, Jiang G. 2021.** Expression of PD-L1 and BRCA1 in triple-negative breast cancer patients and relationship with clinicopathological characteristics. *Evidence-Based Complementary and Alternative Medicine* **2021**:5314016.

**Jarrard DF, Sarkar S, Shi Y, Yeager TR, Magrane G, Kinoshita H, Nassif N, Meisner L, Newton MA, Waldman FM, Reznikoff CA. 1999.** p16/pRb pathway alterations are required for bypassing senescence in human prostate epithelial cells. *Cancer Research* **59(12)**:2957–2964.

**Jiang P, Gu S, Pan D, Fu J, Sahu A, Hu X, Li Z, Traugh N, Bu X, Li B, Liu J, Freeman GJ, Brown MA, Wucherpfennig KW, Liu XS. 2018.** Signatures of T cell dysfunction and exclusion predict cancer immunotherapy response. *Nature Medicine* **24(10)**:1550–1558 DOI 10.1038/s41591-018-0136-1.

**Jiang YZ, Ma D, Suo C, Shi J, Xue M, Hu X, Xiao Y, Yu KD, Liu YR, Yu Y, Zheng Y, Li X, Zhang C, Hu P, Zhang J, Hua Q, Zhang J, Hou W, Ren L, Bao D, Li B, Yang J, Yao L, Zuo WJ, Zhao S, Gong Y, Ren YX, Zhao YX, Yang YS, Niu Z, Cao ZG, Stover DG, Verschraegen C, Kaklamani V, Daemen A, Benson JR, Takabe K, Bai F, Li DQ, Wang P, Shi L, Huang W, Shao ZM. 2019.** Genomic and transcriptomic landscape of triple-negative breast cancers: subtypes and treatment strategies. *Cancer Cell* **35(3)**:428–440.e5 DOI 10.1016/j.ccell.2019.02.001.

**Kollareddy M, Martinez LA. 2021.** Distinct classes of flavonoids and epigallocatechin gallate, polyphenol affects an oncogenic mutant p53 protein, cell growth and invasion in a tnbc breast cancer cell line. *Cells* **10(4)**:797.

**Krtolica A, Parrinello S, Lockett S, Desprez PY, Campisi J. 2001.** Senescent fibroblasts promote epithelial cell growth and tumorigenesis: a link between cancer and aging. *Proceedings of the National Academy of Sciences of the United States of America* **98(21)**:12072–12077.

**Krumpel M, Reithmeier A, Senge T, Baeumler TA, Frank M, Nyholm PG, Ek-Rylander B, Andersson G. 2015.** The small chemical enzyme inhibitor 5-phenylnicotinic acid/CD13 inhibits cell migration and invasion of tartrate-resistant acid phosphatase/ACP5-overexpressing MDA-MB-231 breast cancer cells. *Experimental Cell Research* **339(1)**:154–162 DOI 10.1016/j.yexcr.2015.09.019.

**Lasry A, Ben-Neriah Y. 2015.** Senescence-associated inflammatory responses: aging and cancer perspectives. *Trends in Immunology* **36(4)**:217–228 DOI 10.1016/j.it.2015.02.009.

**Lehmann BD, Bauer JA, Chen X, Sanders ME, Chakravarthy AB, Shyr Y, Pietenpol JA. 2011.** Identification of human triple-negative breast cancer subtypes and preclinical models for selection of targeted therapies. *Journal of Clinical Investigation* **121(7)**:2750–2767 DOI 10.1172/JCI45014.

**Li A, Keck JM, Parmar S, Patterson J, Labrie M, Creason AL, Johnson BE, Downey M, Thomas G, Beadling C, Heiser LM, Kolodzie A, Guimaraes AR, Corless CL, Gray JW, Mills GB, Bergan RC, Mitri ZI. 2021.** Characterizing advanced breast cancer heterogeneity and treatment resistance through serial biopsies and comprehensive analytics. *NPJ Precision Oncology* **5(1)**:28 DOI 10.1038/s41698-021-00165-4.

**Li S, Wei X, He J, Tian X, Yuan S, Sun L. 2018.** Plasminogen activator inhibitor-1 in cancer research. *Biomedicine & Pharmacotherapy* **105**:83–94 DOI 10.1016/j.biopha.2018.05.119.

**Li T, Fu J, Zeng Z, Cohen D, Li J, Chen Q, Li B, Liu XS. 2020.** TIMER2.0 for analysis of tumor-infiltrating immune cells. *Nucleic Acids Research* **48(W1)**:W509–W514 DOI 10.1093/nar/gkaa407.

**Liberzon A, Birger C, Thorvaldsdóttir H, Ghandi M, Mesirov JP, Tamayo P. 2015.** The Molecular Signatures Database (MSigDB) hallmark gene set collection. *Cell Systems* **1(6)**:417–425 DOI 10.1016/j.cels.2015.12.004.

**Lin W, Wang X, Wang Z, Shao F, Yang Y, Cao Z, Feng X, Gao Y, He J. 2021.** Comprehensive analysis uncovers prognostic and immunogenic characteristics of cellular senescence for lung adenocarcinoma. *Frontiers in Cell and Developmental Biology* **9**:780461 DOI 10.3389/fcell.2021.780461.

**Liu D, Hornsby PJ. 2007.** Senescent human fibroblasts increase the early growth of xenograft tumors via matrix metalloproteinase secretion. *Cancer Research* **67(7)**:3117–3126 DOI 10.1158/0008-5472.CAN-06-3452.

**Malorni L, Shetty PB, De Angelis C, Hilsenbeck S, Rimawi MF, Elledge R, Osborne CK, De Placido S, Arpino G. 2012.** Clinical and biologic features of triple-negative breast cancers in a large cohort of patients with long-term follow-up. *Breast Cancer Research and Treatment* **136(3)**:795–804 DOI 10.1007/s10549-012-2315-y.

**Masiero M, Simões FC, Han HD, Snell C, Peterkin T, Bridges E, Mangala LS, Wu SY, Pradeep S, Li D, Han C, Dalton H, Lopez-Berestein G, Tuynman JB, Mortensen**

N, Li JL, Patient R, Sood AK, Banham AH, Harris AL, Buffa FM. 2013. A core human primary tumor angiogenesis signature identifies the endothelial orphan receptor ELTD1 as a key regulator of angiogenesis. *Cancer Cell* **24(2)**:229–241 DOI 10.1016/j.ccr.2013.06.004.

**Mayakonda A, Lin DC, Assenov Y, Plass C, Koeffler HP. 2018.** Maftools: efficient and comprehensive analysis of somatic variants in cancer. *Genome Research* **28(11)**:1747–1756 DOI 10.1101/gr.239244.118.

**Mermel CH, Schumacher SE, Hill B, Meyerson ML, Beroukhim R, Getz G. 2011.** GISTIC2.0 facilitates sensitive and confident localization of the targets of focal somatic copy-number alteration in human cancers. *Genome Biology* **12(4)**:R41 DOI 10.1186/gb-2011-12-4-r41.

**Metzger-Filho O, Tutt A, Azambuja Ede, Saini KS, Viale G, Loi S, Bradbury I, Bliss JM, Jr HAAzim, Ellis P, Leo ADi, Baselga J, Sotiriou C, Piccart-Gebhart M. 2012.** Dissecting the heterogeneity of triple-negative breast cancer. *Journal of Clinical Oncology* **30(15)**:1879–1887 DOI 10.1200/JCO.2011.38.2010.

**Nik-Zainal S, Davies H, Staaf J, Ramakrishna M, Glodzik D, Zou X, Martincorena I, Alexandrov LB, Martin S, Wedge DC, Van Loo P, Ju YS, Smid M, Brinkman AB, Morganella S, Aure MR, Lingjærde OC, Langerød A, Ringnér M, Ahn SM, Boyault S, Brock JE, Broeks A, Butler A, Desmedt C, Dirix L, Dronov S, Fatima A, Foekens JA, Gerstung M, Hooijer GK, Jang SJ, Jones DR, Kim HY, King TA, Krishnamurthy S, Lee HJ, Lee JY, Li Y, McLaren S, Menzies A, Mustonen V, O'Meara S, Pauporté I, Pivot X, Purdie CA, Raine K, Ramakrishnan K, Rodríguez-González FG, Romieu G, Sieuwerts AM, Simpson PT, Shepherd R, Stebbings L, Stefansson OA, Teague J, Tommasi S, Treilleux I, Eynden GGVanden, Vermeulen P, Vincent-Salomon A, Yates L, Caldas C, Van't Veer L, Tutt A, Knappskog S, Tan BK, Jonkers J, Borg Å, Ueno NT, Sotiriou C, Viari A, Futreal PA, Campbell PJ, Span PN, Van Laere S, Lakhani SR, Eyfjord JE, Thompson AM, Birney E, Stunnenberg HG, Van de Vijver MJ, Martens JW, Børresen-Dale AL, Richardson AL, Kong G, Thomas G, Stratton MR. 2016.** Landscape of somatic mutations in 560 breast cancer whole-genome sequences. *Nature* **534(7605)**:47–54 DOI 10.1038/nature17676.

**Pareja F, Reis-Filho JS. 2018.** Triple-negative breast cancers—a panoply of cancer types. *Nature Reviews Clinical Oncology* **15(6)**:347–348 DOI 10.1038/s41571-018-0001-7.

**Ritchie ME, Phipson B, Wu D, Hu Y, Law CW, Shi W, Smyth GK. 2015.** limma powers differential expression analyses for RNA-sequencing and microarray studies. *Nucleic Acids Research* **43(7)**:e47 DOI 10.1093/nar/gkv007.

**Sanchez-Vega F, Mina M, Armenia J, Chatila WK, Luna A, La KC, Dimitriadoy S, Liu DL, Kantheti HS, Saghafinia S, Chakravarty D, Daian F, Gao Q, Bailey MH, Liang WW, Foltz SM, Shmulevich I, Ding L, Heins Z, Ochoa A, Gross B, Gao J, Zhang H, Kundra R, Kandoth C, Bahceci I, Dervishi L, Dogrusoz U, Zhou W, Shen H, Laird PW, Way GP, Greene CS, Liang H, Xiao Y, Wang C, Iavarone A, Berger AH, Bivona TG, Lazar AJ, Hammer GD, Giordano T, Kwong LN, McArthur G, Huang C, Tward AD, Frederick MJ, McCormick F, Meyerson M, Allen EMVan, Cherniack**
AD, Ciriello G, Sander C, Schultz N. 2018. Oncogenic signaling pathways in the cancer genome atlas. *Cell* **173(2)**:321–337.e10 DOI 10.1016/j.cell.2018.03.035.

Seker F, Cingoz A, Sur-Erdem İ, Erguder N, Erkent A, Uyulur F, Esai Selvan M, Gümüş ZH, Gönen M, Bayraktar H, Wakimoto H, Bagci-Onder T. 2019. Identification of SERPINE1 as a regulator of glioblastoma cell dispersal with transcriptome profiling. *Cancers* **11(11)**:1651.

Shang C, Xu D. 2022. Epidemiology of breast cancer. *Oncologie* **24(4)**:649–663 DOI 10.32604/oncologie.2022.027640.

Shen W, Song Z, Xiao Z, Huang M, Shen D, Gao P, Qian X, Wang M, He X, Wang T, Li S, Song X. 2022. Sangerbox: a comprehensive, interaction-friendly clinical bioinformatics analysis platform. *iMeta* **1(3)** DOI 10.1002/imt2.36.

Subramanian A, Tamayo P, Mootha VK, Mukherjee S, Ebert BL, Gillette MA, Paulovich A, Pomeroy SL, Golub TR, Lander ES, Mesirov JP. 2005. Gene set enrichment analysis: a knowledge-based approach for interpreting genome-wide expression profiles. *Proceedings of the National Academy of Sciences of the United States of America* **102(43)**:15545–15550.

Takasugi M, Yoshida Y, Ohtani N. 2022. Cellular senescence and the tumour microenvironment. *Molecular Oncology* **16(18)**:3333–3351 DOI 10.1002/1878-0261.13268.

Therneau T, Hart S, Kocher J. 2023. Calculating samplesSize estimates for RNA Seq studies. R package version 1.42.0..

Tong L, Yu X, Wang S, Chen L, Wu Y. 2023. Research progress on molecular subtyping and modern treatment of triple-negative breast cancer. *Breast Cancer* **15**:647–658.

Wang L, Lankhorst L, Bernards R. 2022. Exploiting senescence for the treatment of cancer. *Nature Reviews Cancer* **22(6)**:340–355 DOI 10.1038/s41568-022-00450-9.

Wilkerson MD, Hayes DN. 2010. ConsensusClusterPlus: a class discovery tool with confidence assessments and item tracking. *Bioinformatics* **26(12)**:1572–1573 DOI 10.1093/bioinformatics/btq170.

Wu X, Tao P, Zhou Q, Li J, Yu Z, Wang X, Li J, Li C, Yan M, Zhu Z, Liu B, Su L. 2017. IL-6 secreted by cancer-associated fibroblasts promotes epithelial-mesenchymal transition and metastasis of gastric cancer via JAK2/STAT3 signaling pathway. *Oncotarget* **8(13)**:20741–20750 DOI 10.18632/oncotarget.15119.

Xiao Y, Ma D, Zhao S, Suo C, Shi J, Xue MZ, Ruan M, Wang H, Zhao J, Li Q, Wang P, Shi L, Yang WT, Huang W, Hu X, Yu KD, Huang S, Bertucci F, Jiang YZ, Shao ZM. 2019. Multi-omics profiling reveals distinct microenvironment characterization and suggests immune escape mechanisms of triple-negative breast cancer. *Clinical Cancer Research* **25(16)**:5002–5014 DOI 10.1158/1078-0432.CCR-18-3524.

Xue W, Zender L, Miething C, Dickins RA, Hernando E, Krizhanovsky V, Cordon-Cardo C, Lowe SW. 2007. Senescence and tumour clearance is triggered by p53 restoration in murine liver carcinomas. *Nature* **445(7128)**:656–660 DOI 10.1038/nature05529.

Yang JD, Ma L, Zhu Z. 2019. SERPINE1 as a cancer-promoting gene in gastric adenocarcinoma: facilitates tumour cell proliferation, migration, and invasion by regulating

EMT. *Journal of Chemotherapy* **31**(**7-8**):408–418
DOI 10.1080/1120009X.2019.1687996.

**Yasuda T, Koiwa M, Yonemura A, Miyake K, Kariya R, Kubota S, Yokomizo-Nakano T, Yasuda-Yoshihara N, Uchihara T, Itoyama R, Bu L, Fu L, Arima K, Izumi D, Iwagami S, Eto K, Iwatsuki M, Baba Y, Yoshida N, Ohguchi H, Okada S, Matsusaki K, Sashida G, Takahashi A, Tan P, Baba H, Ishimoto T. 2021.** Inflammation-driven senescence-associated secretory phenotype in cancer-associated fibroblasts enhances peritoneal dissemination. *Cell Reports* **34**(**8**):108779 DOI 10.1016/j.celrep.2021.108779.

**Yoshihara K, Shahmoradgoli M, Martínez E, Vegesna R, Kim H, Torres-Garcia W, Treviño V, Shen H, Laird PW, Levine DA, Carter SL, Getz G, Stemke-Hale K, Mills GB, Verhaak RG. 2013.** Inferring tumour purity and stromal and immune cell admixture from expression data. *Nature Communications* **4**:2612 DOI 10.1038/ncomms3612.

**You D, Jeong Y, Yoon SY, Kim SA, Lo E, Kim SW, Lee JE, Nam SJ, Kim S. 2021.** Entelon(Ⓡ) (vitis vinifera seed extract) prevents cancer metastasis via the down-regulation of interleukin-1 alpha in triple-negative breast cancer cells. *Molecules* **26**(**12**):3644.

**Yu KD, Zhu R, Zhan M, Rodriguez AA, Yang W, Wong S, Makris A, Lehmann BD, Chen X, Mayer I, Pietenpol JA, Shao ZM, Symmans WF, Chang JC. 2013.** Identification of prognosis-relevant subgroups in patients with chemoresistant triple-negative breast cancer. *Clinical Cancer Research* **19**(**10**):2723–2733 DOI 10.1158/1078-0432.CCR-12-2986.

**Yu TJ, Ma D, Liu YY, Xiao Y, Gong Y, Jiang YZ, Shao ZM, Hu X, Di GH. 2021.** Bulk and single-cell transcriptome profiling reveal the metabolic heterogeneity in human breast cancers. *Molecular Therapy Molecular Therapy* **29**(**7**):2350–2365 DOI 10.1016/j.ymthe.2021.03.003.

**Yue T, Chen S, Zhu J, Guo S, Huang Z, Wang P, Zuo S, Liu Y. 2021.** The aging-related risk signature in colorectal cancer. *Aging* **13**(**5**):7330–7349 DOI 10.18632/aging.202589.

**Zhang Q, Lei L, Jing D. 2020a.** Enhancer transcription reveals subtype-specific gene expression programs controlling breast cancer pathogenesis. *Genome Research* **28**(**2**):159–170 DOI 10.1101/gr.226019.117.

**Zhang Q, Lei L, Jing D. 2020b.** Knockdown of SERPINE1 reverses resistance of triple-negative breast cancer to paclitaxel via suppression of VEGFA. *Oncology Reports* **44**(**5**):1875–1884 DOI 10.3892/or.2020.7770.

**Zhang Y, Wang S, Liu Z, Yang L, Liu J, Xiu M. 2019.** Increased Six1 expression in macrophages promotes hepatocellular carcinoma growth and invasion by regulating MMP-9. *Journal of Cellular and Molecular Medicine* **23**(**7**):4523–4533 DOI 10.1111/jcmm.14342.

**Zhong G, Lou W, Shen Q, Yu K, Zheng Y. 2020.** Identification of key genes as potential biomarkers for triple-negative breast cancer using integrating genomics analysis. *Molecular Medicine Reports* **21**(**2**):557–566 DOI 10.3892/mmr.2019.10867.

**Zhou P, Zheng G, Li Y, Wu D, Chen Y. 2020.** Construction of a circRNA-miRNA-mRNA network related to macrophage infiltration in hepatocellular carcinoma. *Frontiers in Genetics* **11**:1026 DOI 10.3389/fgene.2020.01026.

**Zhu D, Yang J, Xu J. 2022.** G-protein-coupled estrogen receptor enhances the stemness of triple-negative breast cancer cells and promotes malignant characteristics. *Oncologie* **24(3)**:471–482 DOI 10.32604/oncologie.2022.024062.