# Peer review of "Research and experimental verification on the mechanisms of cellular senescence in triple-negative breast cancer"

_PeerJ, doi:10.7717/peerj.16935_

## Round 0.1 · original submission · Major Revisions

Both reviewers raised issues that need to be addressed from different perspectives on the manuscript. Please modify or reply point by point according to the reviewer's requirements or suggestions.

Reviewer 1 ·

Basic reporting

no comment

Experimental design

no comment

Validity of the findings

no comment

Additional comments

Triple-negative breast cancer (TNBC) is an aggressive subtype characterized by extensive intratumoral heterogeneity. In this study, the authors comprehensively revealed the mechanisms of cellular senescence in TNBC based on single-cell data and utilizing multiple bioinformatics tools. Overall, this is a comprehensive study. However, the content in the manuscript needs to be revised accordingly to be able to better be published in a journal.
1. This study focuses on the mechanism of TNBC cellular senescence. Therefore, in the "Background" section of the abstract, the authors should add information about the significance of TNBC cell senescence or the purpose of investigating the mechanism of TNBC cell senescence.
2. Please note that in the summary section, for the first occurrence of an acronym, please provide its full name, e.g., TCGA, ESTIMATE, LASSO, etc.
3. In line 23, it is proposed to add means of access to relevant pathways related to cellular senescence.
4. The authors performed some cellular experiments such as “qRT-PCR” and “Transwell assay”, however, these are not represented in the abstract section.
5. The references used in the manuscript as a whole are too dated, and the authors should update the references in this manuscript appropriately.
6. In line 113, it is suggested to add some descriptions that have a description of the significance of the study, such as "In conclusion, our study provides a new approach to the diagnosis and treatment of patients with TNBC."
7. At line 262, the author needs to summarize the section in a concise manner that will enable the reader to clearly understand what kind of information has been gained through the analysis of the section.
8. Figure 5C clearly shows the gene amplifications and deletions in the three cell subtypes, especially in Figure 5C, however the authors did not describe the corresponding chromosomal changes regarding CNV in the results section.
9. For a complete description of the drug sensitivity analysis in Figure 10F. The authors should make a brief list of the drugs analyzed and describe the significant differences between the high and low risk groups.
10. Please note that the clarity of the image meets the criteria for publication.
11. In line 455, "TNBC shows a high heterogeneity" is suggested to be changed to "TNBC is a highly heterogeneous type of cancer."
12. At lines 464-471, the authors merely list the relevant genes, which would be meaningless.
13. In line 472-499, it is only in the repetition of writing the results section, which is not necessary. The authors should have based their results on previous studies with subtype classification of TNBC or on subtype classification of cellular senescence mechanisms as a comparison.
14. The conclusion section (line 537) is suggested to be rewritten and instead of repeating the content of the results, the authors should present the key conclusions and reveal the importance of what the study brings to the diagnosis, prognosis and treatment of TNBC patients.

Reviewer 2 ·

Basic reporting

This study focuses on revealing the cellular senescence mechanism of triple-negative breast cancer (TNBC) through both bioinformatics analysis and cellular experiments. In this study, we first obtained single-cell transcriptome data of TNBC from public databases, and clustered the data into different cellular subpopulations by bioinformatics to explore the cellular subpopulations associated with the pathogenesis of TNBC. Next, a cellular senescence pathway-related score was calculated in combination with ssGSEA, and the samples were categorized according to this score, and the differences in immune characteristics of the samples between groups were assessed to reveal the variability between samples in terms of immune cell infiltration, and benefit of immunotherapy. Subsequently, the study constructed a prognostic model based on TNBC cell senescence-related genes, and verified the regulatory role of the model characterized genes in the malignant phenotype of TNBC cell lines by cellular experiments. In conclusion, the study is overall rigorous and meets the requirements for publication
1. It is suggested that the Introduction section explains what challenges TNBC heterogeneity poses to clinical cancer treatment and what difficulties it causes in the field of immunotherapy or targeted therapies, thus sharpening the intent of this study.
2. Cellular senescence is the main entry point of the study in this paper, thus the Introduction section should explain the relevant literature in more specific and detailed, such as what factors affect cellular senescence, what are the special features of cellular senescence in TNBC that differentiate it from other cancer types in the existing studies, and what are the existing therapies for cellular senescence in TNBC? Please clarify.
3. Perhaps senescence-associated genes have not been explored in TNBC, but have cellular senescence-associated genes or some of the cytokines that cause cellular senescence with TNBC been elucidated in existing reports? It is suggested to add a description of the relevant literature thus making the whole text more logical.

Experimental design

4. Why did this study use only Univariate Cox and LASSO regression analyses for this aspect of screening for TNBC prognosis-related genes, and did more in-depth regression analyses after this leading to a more specific and definitive presentation of the results? Please provide a brief rationale and add analyses as necessary.
5. Differences in the level of immune cell infiltration in the immune microenvironment should be an important aspect of this paper to reveal the mechanisms associated with TNBC, but is the present study slightly insufficient to reveal this using only a single method? It is recommended to add relevant analyses when necessary.
6. In this study, two genes, CD24 and KRT19, which are highly expressed in epithelial cells, were excavated after single-cell transcriptome analysis, but the later study did not focus on these two genes, should they be included in the TNBC prognosis-related genes for subsequent studies? Please give a reasonable explanation.

Validity of the findings

7. It is recommended that the AUC values of the ROC curves be listed in detail in the description of the results in Figure 8, and that the AUC values close to 0.7 be explained as to what they indicate.

Additional comments

8. Please provide a brief overview of this study in the Discussion section, highlighting which analyses led to which main results, while the discussion of prognosis-related genes is too brief, and it is recommended that it be supplemented, in particular highlighting the mechanisms by which these genes are associated with cellular senescence.
9. The three subgroups of subtypes in this study differed in terms of TME immune cell infiltration levels, immune checkpoint gene expression levels, immunotherapy response and gene mutations, and thus it is suggested that the original article focus on these aspects, reasonably hypothesize possible mechanisms affecting the progression of TNBCs, and correlate these aspects with the characteristics of TNBC cellular senescence. It is suggested that this focus be used to reorganize the content of lines 472-499.
10. Since this paper is a combination of bioinformatics analysis and cellular experiments to illustrate the mechanisms related to TNBC, it should specify what kind of results were obtained from the cellular experiments, such as promoting the malignant phenotype of the cells and thus affecting the cancer progression, etc., instead of skimming over the results of the study, it is suggested to add in Abstract, Introduction, and Conclusion.

---

## Round 0.2 · accepted · Accept

Both reviewers have expressed their appreciation for the thoroughness and clarity of your research. Based on the positive feedback from the reviewers, we are delighted to accept your manuscript for publication in PeerJ. We believe that your research will make a substantial contribution to the field and will be of great interest to our readership. We would like to express our gratitude for your contribution to scientific research and for choosing PeerJ as the platform to share your findings. We look forward to receiving the revised manuscript and working with you towards the successful publication of your research.

Reviewer 1 ·

Basic reporting

no comment

Experimental design

no comment

Validity of the findings

no comment

Additional comments

This manuscript conducted a comprehensive study on triple negative breast cancer (TNBC), focusing on the role of cell aging. Advanced methodologies such as single-cell RNA sequencing (scRNA seq) analysis, gene set enrichment analysis (ssGSEA), and consensus clustering were employed to identify molecular clusters. It also includes using various bioinformatics tools for immune score calculation and constructing risk models based on gene expression data. Three subtypes of TNBC were identified based on pathways related to cellular aging, exhibiting different prognoses and levels of immune infiltration. It also explored the immune escape potential of certain subtypes and validated the TNBC risk model through qRT PCR and Transwell experiments.
Overall, this manuscript provides important insights into the molecular mechanisms of TNBC, particularly its role in cellular aging, and proposes potential prognostic factors and therapeutic targets.

Reviewer 2 ·

Basic reporting

1、The language of the manuscript is clear and accurate, and the use of professional terminology is appropriate and easy to understand. When explaining complex scientific concepts, the author is able to maintain both scientificity and ease of understanding, which is of great guiding significance for both professional and non professional readers.
2、The article has a rigorous structure and follows the conventional format of scientific research papers. From the introduction to the methodology, to the results and discussion, each section is well-organized and logically clear. Especially in the methods and results section, the organization of information enables readers to smoothly follow the research process and understand every step of the research.
3、In the background part, the author gives a detailed description of the characteristics, current research status and existing problems of triple negative breast cancer, which provides a solid foundation for the importance and necessity of this study. In addition, the exploration of the role of cellular aging in TNBC not only demonstrates the author's in-depth understanding of the field, but also provides support for the unique perspective of this study.

Experimental design

no comment

Validity of the findings

The manuscript not only relies on bioinformatics analysis, but also includes laboratory validation, such as quantitative reverse transcription polymerase chain reaction (qRT PCR) and Transwell experiments, further enhancing the credibility of research conclusions. This verification method from theory to practice demonstrates the comprehensiveness of the research, ensuring that the research findings not only remain at the theoretical level, but also have experimental support.

Additional comments

The manuscript shows some innovation in exploring the role of cell aging in triple negative breast cancer (TNBC). Through in-depth research on the relationship between cellular aging and TNBC, this manuscript provides a new perspective for understanding the molecular mechanisms of TNBC, particularly in terms of cellular aging pathways and their impact on the development of TNBC. This research direction not only fills the gap in existing research, but may also have far-reaching implications for future treatment strategies.